# Melanometry for objective evaluation of skin pigmentation in pulse oximetry studies
**Sandhya Vasudevan** ✉**, William C. Vogt, Sandy Weininger & T. Joshua Pfefer**

Pulse oximetry enables real-time, noninvasive monitoring of arterial blood oxygen levels. However, results can vary with skin color, thus detecting disparities during clinical validation studies requires an accurate measure of skin pigmentation. Recent clinical studies have used subjective methods such as self-reported color, race/ethnicity to categorize skin. Melanometers based on optical reflectance may offer a more effective, objective approach to assess pigmentation. Here, we review melanometry approaches and assess evidence supporting their use as clinical research tools. We compare performance data, including repeatability, robustness to confounders, and compare devices to each other, to subjective methods, and high-quality references. Finally, we propose best practices for evaluating melanometers and discuss alternate optical approaches that may improve accuracy. Whilst evidence indicates that melanometers can provide superior performance to subjective approaches, we encourage additional research and standardization efforts, as these are needed to ensure consistent and reliable results in clinical studies.

An important indicator of health is the percentage of hemoglobin molecules that are bound to oxygen in arterial blood ($SaO_2$). Pulse oximetry is a commonly used medical technology that generates a value ($SpO_2$) which is an estimate of $SaO_2$ based on noninvasive optical measurements. $SpO_2$ measurements are used to inform healthcare decisions in a broad range of clinical settings. However, evidence of $SpO_2$ measurement disparities correlated with biological factors such as skin color have been documented[1,2]. The significance of these disparities was highlighted during the COVID-19 pandemic when it was found that Black patients were three times more likely than white patients to be diagnosed as having normal $SpO_2$ (normoxemic $SpO_2$) despite having a low $SaO_2$ level (occult hypoxemia)[3,4]. This was found to have impacts on health care, including delayed detection of COVID-19 infection[5].

As part of the response to the impact of these clinically significant racial disparities highlighted during the COVID-19 pandemic, the U.S. Food and Drug Administration (FDA) released a safety communication raising concerns about the accuracy of pulse oximetry measurements across racial groups, and highlighting the importance of understanding and resolving this problem[6]. In 2022, FDA held a panel meeting to gather input from experts[7]. While the mechanisms underlying racial disparities in pulse oximeter performance remain unclear, panel members recognized a need for accurate assessments of skin pigmentation in clinical studies, including the potential use of objective approaches[8].

This review provides an introduction to melanometry as a potential tool for objective assessment of skin pigmentation, to facilitate assessments of the accuracy of pulse oximetry and other clinical optical sensing devices. Melanometers are devices that use reflected light at visible and near-infrared wavelengths to provide objective, quantitative measurements of pigmentation. Prior reviews of melanometry compared different approaches[9–13] or focused on clinical feasibility for specific dermatology applications such as monitoring anti-scarring therapies[14,15]. Here we quantitatively analyze melanometer performance from published data. Specifically, we evaluate performance attributes including comparison to subjective evaluation methods, inter-melanometer agreement, repeatability, and accuracy versus the established best approaches (gold standards). Finally, we discuss the potential for melanometer best practices, standardization, and alternate approaches warranting future study and development.

## The importance of accurately measuring melanin

Melanin is a complex polymer produced in various places in the human body, including the skin. It is a dominant biological chromophore (absorber) in the ultraviolet (UV), visible, and near-infrared spectral regions[16–18]. As a consequence, it is a major contributor to differences in skin color (pigmentation). Melanin can impact a variety of biophotonic technologies that require measurement of light that has propagated through skin. Such devices include spectroscopic devices, for example, pulse oximeters, as well

Center for Devices and Radiological Health, Food and Drug Administration, 10903 New Hampshire Ave, Silver Spring, MD 20993, USA.
✉e-mail: sandhya.vasudevan@fda.hhs.gov

as non-pulsatile regional tissue oximeters that are used for measurements in the skin and brain[19–21]. Researchers have used a variety of approaches to categorize patients whilst evaluating the impact of skin pigmentation on the performance of optical devices. Skin has been categorized as light (white), dark (African American), or intermediate (others)[1,2] or using classification methods such as the Fitzpatrick skin phototype (FSP)[22–26], Munsell color system[27–30] or Massey Skin Color Score[31,32]. Because these methods do not measure actual melanin content, they have been criticized for being subjective and susceptible to inter-operator variability[33,34]. More recent studies of pulse oximeters have used self-identified race[3,5,35,36] or subjective evaluation of race/ethnic origin[4,37–55] to categorize participants. However, given that many people have mixed ethnicities and that skin pigmentation levels can vary within any ethnic group, conflation of ethnicity with skin pigmentation may also produce misleading results and not permit accurate investigation of any oximeter bias[33]. Some reports have acknowledged limitations of these subjective approaches[4,5].

There is growing support from researchers[34,53,56] and professional societies[57,58] to address the need for improved methods by implementing objective optical methods to quantify melanin content. It is hoped that by measuring skin pigmentation as a continuous variable, rather than grouping skin tones into discrete subjective categories (e.g., light, intermediate, dark), it will be possible to enroll study participants that more accurately represent the range of skin pigmentation levels necessary. Furthermore, accurate melanin content measurements should provide a higher quality metric for statistical analysis of trends or differences in clinical results. Such data may also be useful for optimizing device algorithms and executing numerical models of light-tissue interactions that enhance quantitative understanding of the role of melanin in optical devices[59–61].

Over the past 20 years, some scientific fields have widely adopted melanometers as research tools. While commercially available, the vast majority of these devices are not cleared or approved by the FDA as medical devices. In dermatology, melanometers have been implemented for studying vitiligo[62,63], scar tissue[14,64], melasma[65], and psoriasis[66,67]. Anthropologists use melanometers to measure skin pigmentation in individuals of different ancestry[68–70]. By using techniques that have been successfully implemented in these fields, it may be possible to substantially improve the rigor and quality of clinical studies on optical device performance.

## Tissue optics of skin pigmentation

Light-tissue interactions determine visualized skin color, as well as the optical signals detected by pulse oximeters and melanometers. While light propagation across heterogeneous material such as skin depends on many factors, the most significant variables are the wavelength-dependent optical properties and spatial distribution of key chromophores, such as melanin and oxygenated/deoxygenated hemoglobin. In addition to absorption, light is scattered by tissue microstructures such as cells and collagen fibers, resulting in high levels of diffuse reflectance at the tissue surface. Detected signals are also influenced by illumination parameters (e.g., spectral distribution, intensity, angular distribution) and light collection characteristics (e.g., contact vs. standoff geometry, acceptance angle). Light delivered to the skin first interacts with the epidermis, an avascular structure containing melanin (Fig. 1a, b)[61,71]. The deeper dermis layer is highly vascularized but contains no melanin (Fig. 1a, b)[61,71]. Epidermis and dermis layer thicknesses vary with anatomical site but are typically 50–150 μm and 1–4 mm, respectively.

In a diverse population, variations in melanin content and epidermal microstructure play significant roles in determining visualized skin tone[12,72–75]. Melanocytes located in the basal layer of the epidermis produce melanosomes, which are membrane-bound organelles that synthesize and store melanin[72,74,76]. Epidermal melanin contains multiple pigments including brown or black eumelanin and red or yellow pheomelanin (Fig. 1c)[72,75,77]. Skin color is affected by the total melanin content as well as the size, number, shape, and packaging of melanosomes, although the number of melanocytes tends to be constant for a given anatomic site regardless of skin pigmentation[72,78,79]. A progressive variation in melanosome size with

ethnic or geographic origin has also been revealed, with melanosomes in Black skin being the largest ($1.44 \pm 0.67\ \mu m^2 \times 10^{-2}$) followed by Asian skin ($1.36 \pm 0.15\ \mu m^2 \times 10^{-2}$) and white skin ($0.94 \pm 0.48\ \mu m^2 \times 10^{-2}$)[72,73,75,80]. Melanosomes in white skin are distributed as membrane-bound clusters, whereas melanosomes in Black skin tend to be distributed more individually[72–74] and Asian skin shows a combination of both individual and clustered melanosomes[73]. Black skin tends to contain more eumelanin[72,81] and ~3–6 times more melanin than white skin, whereas Asian skin tends to contain approximately twofold more melanin than white skin[72,74,76,82]. Histological images of the epidermis presented in prior articles often show considerable differences between people with different ethnicities (Fig. 1d)[72]. However, ethnic or geographic categories provide only a moderate degree of correlation with epidermal melanin content[70], and extensive variations in melanin content exists within these groups, including by country[83].

Both melanin and hemoglobin are stronger absorbers for UV and visible wavelengths and weaker for NIR wavelengths[61,84], with melanin exhibiting an exponential decrease with wavelength[16,17,84–86]. Eumelanin and pheomelanin exhibit similar absorption spectra (Fig. 1c) and are often considered as a single chromophore in VIS-NIR (visible and NIR) studies[21,85]. The epidermal tissue absorption can be calculated as follows[87]:

$$\mu_{a,\text{epidermis}}(\lambda) = \left(M_f\mu_{a,\text{mel}}(\lambda) + (1 - M_f)\mu_{a,0}(\lambda)\right)\left(1 - C_{H_2O}\right) + C_{H_2O}\mu_{a,H_2O}(\lambda) \tag{1}$$

where, $M_f$ is the mean volume fraction of melanosomes in the epidermis, $\mu_{a,\text{mel}}$ is the absorption coefficient of a typical melanosome, $\mu_{a,0}$ is the "baseline" absorption coefficient of epidermal tissue without melanin, $C_{H_2O}$ is concentration of water, $\mu_{a,H_2O}$ is the absorption coefficient of water and $\lambda$ is wavelength in nm. The following equations can be used to determine $\mu_{a,\text{mel}}$[16,86] and $\mu_{a,0}$[87,88]:

$$\mu_{a,\text{mel}} = \left(519\ \text{cm}^{-1}\right)\left(\frac{\lambda}{500\text{nm}}\right)^{-3.53} \tag{2}$$

$$\mu_{a,0}(\lambda) = 7.84 \times 10^7\ \lambda^{-3.255} \tag{3}$$

The reduced scattering coefficient of melanin also decreases monotonically with wavelength in the visible wavelength range[89], and correlations between scattering parameters and surface density of melanin pigments indicate that melanin contributes to the overall scattering properties of skin tissue[89,90]. Although scattering may play a role in pulse oximetry racial disparities, given the minimal thickness of the epidermis and relatively small magnitude of change in scattering, the impact of this effect is likely not significant. At typical pulse oximeter wavelengths (660 nm and 940 nm), the absorption coefficient of oxygenated and deoxygenated blood at an average hemoglobin concentration of 150 g/L is much lower than that of highly pigmented epidermis ($M_f = 0.43$, Fig. 2a)[16,91,92].

Clinical measurements across FSP I—VI show increasing melanin content decreases visible and near-infrared reflectance, with stronger changes at shorter wavelengths where melanin absorption is greatest (Fig. 2b)[61,93–95]. Some studies have demonstrated dramatic decreases in reflectance from intermediate to highly pigmented skin[93,95] whereas others have shown more uniform changes[61,94]; this inconsistency is likely due, at least in part, to an imperfect correlation between FSP level and melanin content. Increasing melanin concentration also flattens reflectance spectra in the visible range, weakening features attributed to blood absorption troughs[61,93–95]. The attenuation of UV radiation by melanin in highly pigmented skin reduces the level of UVA and UVB transmission through the epidermis by about 70%[72,96].

## Subjective skin phototype classification methods

Many methods of classifying skin by pigmentation level have been implemented in dermatology, anthropology, and other disciplines. For example, they have been used to evaluate potential for UV photodamage[97,98], evaluate

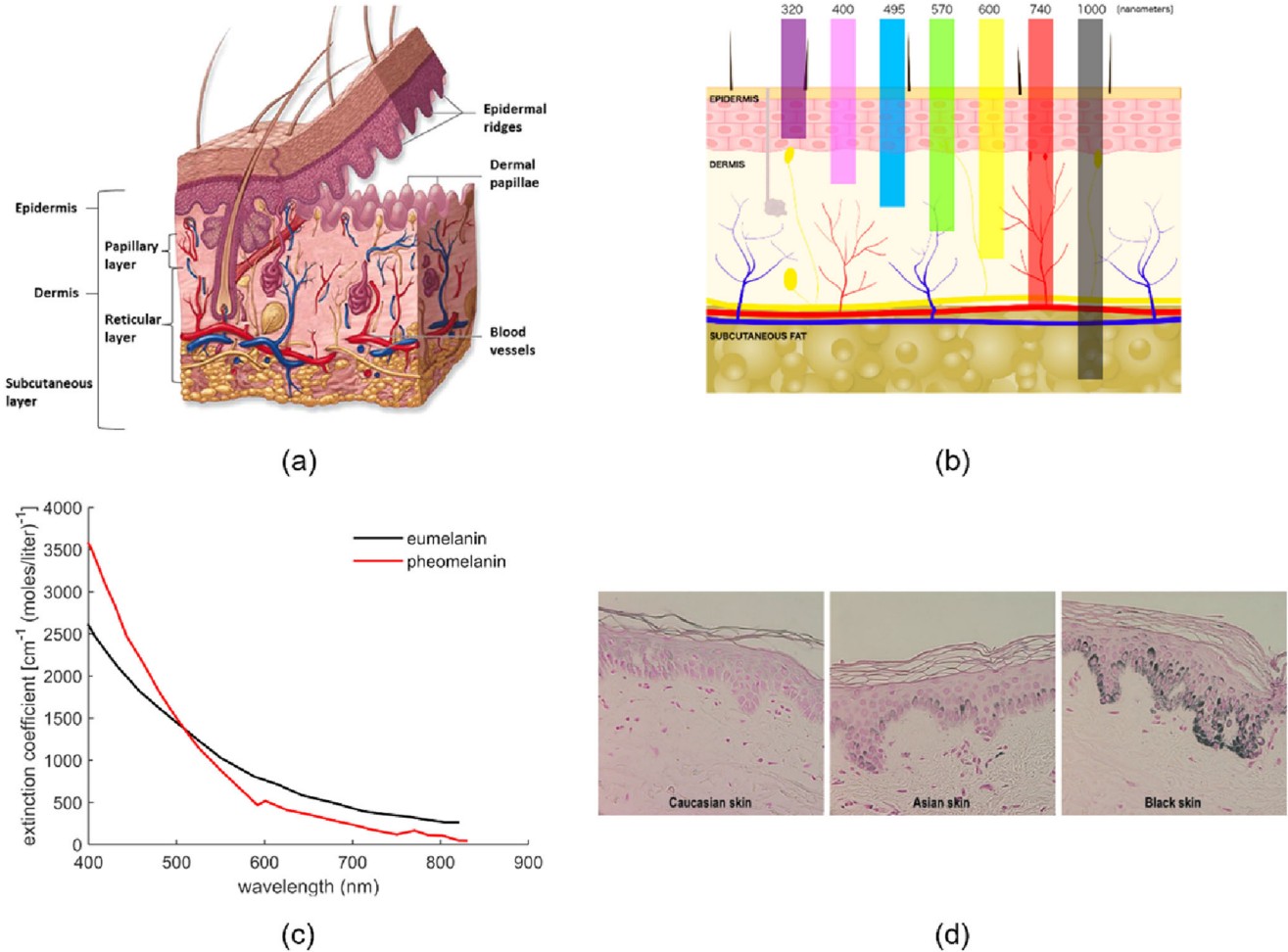

**Fig. 1 | Overview of skin tissue optics.** Schematics illustrating (**a**) skin anatomy[227] (Reprinted with permission from Elsevier) and (**b**) spectral variation in dermal light penetration[228]. **c** Extinction spectra for eumelanin and pheomelanin[229] and (**d**) histology images of skin samples with Fontana-Masson staining indicate variations in epidermal melanin content (epidermis in pink, melanin in black)[72] (Reprinted with permission from Photochem. Photobiol.).

scars and burns[99], and provide a biomarker for genetic studies[69], as well as to study variations in the performance of optical diagnostic devices.

The most ubiquitous subjective classification method in the literature is FSP[22–26], which remains in common medical use. This scale was originally developed in 1975[100] as the Fitzpatrick-Pathak skin typing system to assess ultraviolet light sensitivity in "persons with white skin" to select correct initial UVA dose for patients undergoing oral methoxsalen photo-chemotherapy for psoriasis. This was accomplished via a questionnaire regarding the subject's response to sun exposure, without regard to skin color[100]. Later, FSPs V and VI were added to include "brown and black-skinned persons"[98,101–103]. This approach later became a tool for describing skin color and ethnicity[103,104]. It is worth noting that skin color can refer to constitutive pigmentation – an individual's inherent pigmentation in the absence of solar/ultraviolet exposure, which was originally relevant to FSP – or facultative pigmentation, which accounts for changes due to sun exposure. The current FSP classification denotes six different skin phototypes depending on the individual's erythema sensitivity and ability to tan (see tanning and skin types for FSP in Supplementary Fig. 1a). While perceived FSP has been determined via the use of skin color charts[105,106], there is no established color palette for perceived FSP I-VI categories, and they have not been mapped to a standardized color space. FSP has proven diagnostic and therapeutic value and has been used to predict the risk of photodamage and skin cancer[98], assess the clinical benefits and efficacy of cosmetic procedures[103,107], and has been adopted in FDA guidelines for evaluating sunscreen products[108]. FSP has also been used as a skin color classification tool in a clinical study to support FDA approval of an optical device[109] and in several pulse oximeter performance evaluation studies[22–26].

Von Luschan's chromatic scale (VLS) is a skin color classification method based on a set of 36 opaque, colored glass tiles used as a visual reference (see VLS scale in Supplementary Fig. 1b)[110]. VLS was extensively used in anthropological field research in the 1950s[69], but the legacy of von Luschan and VLS has been considered controversial[111–113]. Other subjective classification methods adopted in literature include self-reported skin color[114–116], racial/ethnic classifications[1–5,35–55], Munsell color chart[27–30], Massey Skin Color Score[31,32], Taylor hyperpigmentation scale[117], and scar assessment scales[14,64,118]. An alternative method that has gained attention recently is the Monk Skin Tone (MST) scale, which is defined by 10 tones and is intended to provide a broader spectrum of pigmentation[119,120]. The MST tones have been mapped to established color spaces, including RGB and Commission Internationale de l'Eclairage (CIE) LAB.

Although subjective methods have been used for categorization of skin pigmentation levels for many years and the aforementioned methods provide a general basis for skin pigment classification, errors can result from observer/user bias, lighting conditions, and skewed self-reporting[11,69,121–125]. Several studies have demonstrated that FSP shows weak correlation with skin color and that physicians predominantly assign non-white individuals to FSP IV, V and VI based on their ethnicity, which has proven to be unreliable[126–128]. Lack of reliability can also be caused when FSP, an approach developed for light-skinned people, is implemented to study a diverse population[129]. Perceived FSP uses a relatively coarse categorization for a

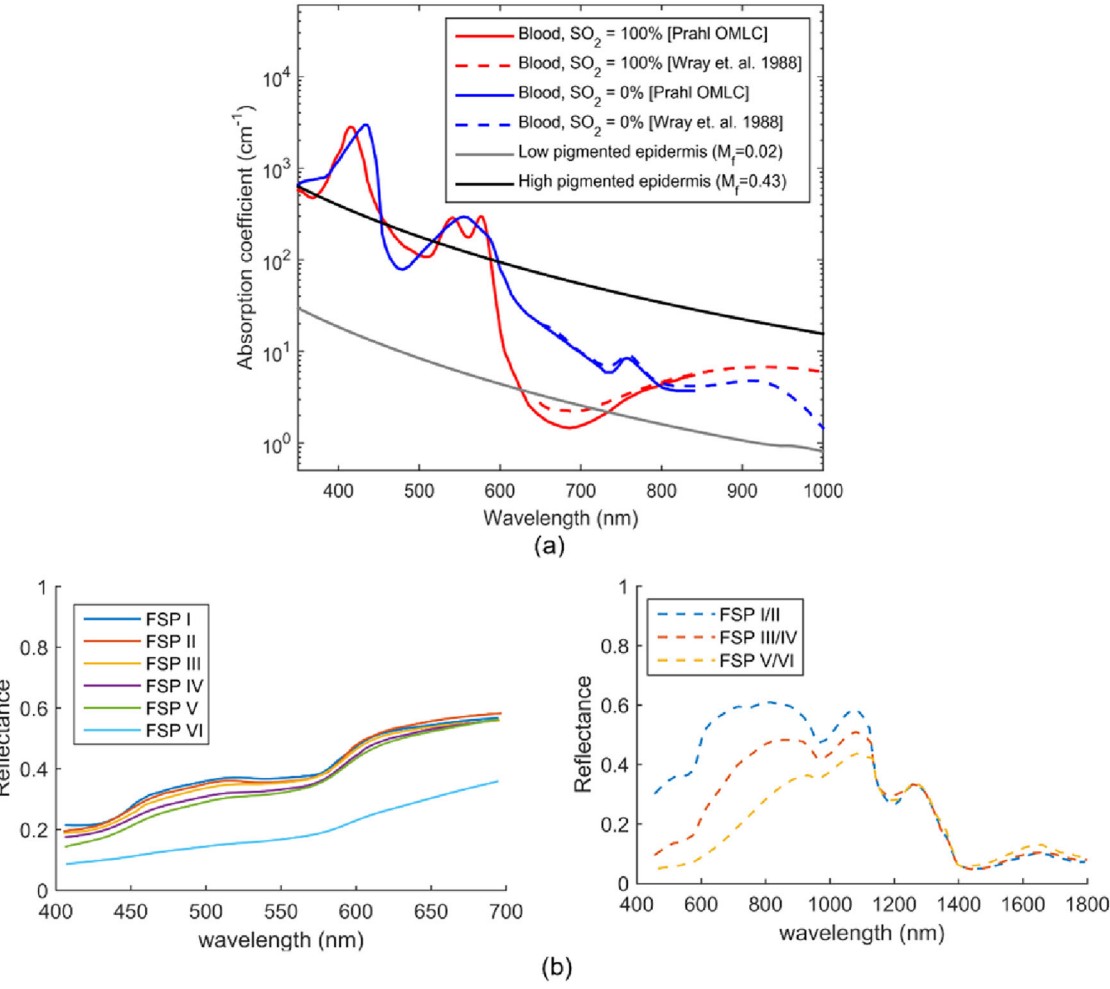

**Fig. 2 | Relationship between tissue optical properties and detected reflectance signals. a** Absorption coefficient spectra for low ($M_f$=0.02, $C_{H_2O}$ = 0.2) and high ($M_f$=0.43, $C_{H_2O}$ = 0.2) pigmented epidermis and oxygenated and deoxygenated blood at 150 g/L hemoglobin concentration[16,87,91,92] and (**b**) reflectance spectra acquired in different studies with participants having FSPs I–VI[93] and FSPs I/II, III/IV, and V/VI[94].

continuum of skin tones (fair to dark)[125] which causes individual categories such as FSP VI to cover a wide range of melanin concentrations (and reflectance values, Fig. 2b). When determined using the original methodology, FSP values do not account for changes in melanin content due to tanning which may impact optical device performance. Moreover, color changes in dark-pigmented skin are generally overlooked by visual assessment[11] and do not address potential intra-individual variations in melanin concentration (immediate pigment darkening or delayed tanning)[125].

## Melanometry principles

Objective, quantitative and observer-independent measures of skin color can be attained by non-invasive devices known as melanometers. Melanometers can be broadly classified based on their acquisition method, processing technique and melanin metrics (Fig. 3a). In this section, we discuss general operating principles as well as the different implemented methodologies for quantifying pigmentation (summarized in Table 1, example devices in Fig. 3b, see also Supplementary Data 1, a version of Table 1 which also includes device parameter settings).

## Data acquisition techniques

Melanometers use visible and near-infrared reflectance measurements. One of the most common approaches involves acquiring narrowband reflectance measurements at selected spectral bands (2–3 wavelengths). The selected wavebands typically include green (relatively strong absorption for melanin), and red and near infrared wavelengths (relatively weak absorption for hemoglobin). To our knowledge, there is only one FDA-cleared narrowband melanometer (Skintel), which was packaged as a component of a light-based skin treatment system[130]. Typically, narrowband melanometers work in direct skin contact, with light emitting diodes (LEDs) illuminating the skin and a silicon photodetector collecting reflected light. Some narrowband devices are calibrated with a provided set of reflecting white and non-reflecting black plates.

A more rigorous and flexible approach to assessing skin pigmentation can be achieved with melanometers using broadband reflectance spectroscopy, which acquire data at many narrow bands, forming a spectrum. This method allows selection of specific wavelengths for data processing algorithms and enables spectral fitting to chromophore absorption signatures, such as oxyhemoglobin (HbO₂), deoxyhemoglobin (HHb), melanin, and water. These systems may use tungsten-halogen lamps, pulsed xenon lamps, or broadband white LEDs depending on the wavelength range of interest, and remitted light is usually detected via a charge-coupled device (CCD) array or silicon photodiode array. Although most reflectance spectroscopic melanometers listed in Table 1 are used in direct skin contact, the Antera 3D and SIAscope II are noncontact devices that acquire large-field 2D maps of skin chromophore concentrations[93,131]. One key benefit of the spectroscopic approach is the flexibility to use spectral data to calculate both a melanin index metric and colorimetric parameters[132,133]. Typically, reflectance spectroscopic devices are calibrated using black and white references[134–136] or a diffuse reflectance standard[85].

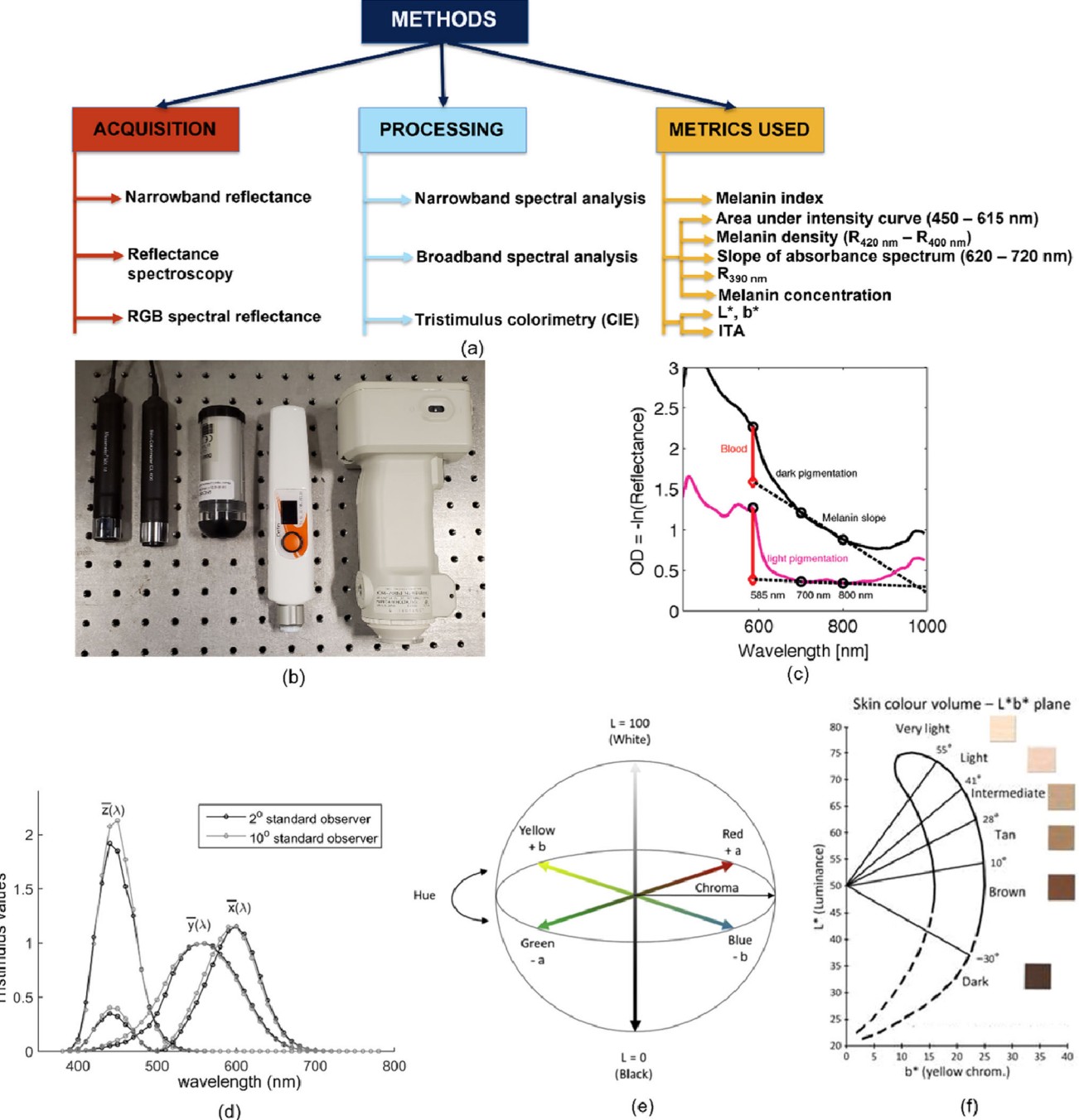

**Fig. 3 | Melanometry methods. a** Overview of acquisition, processing and outputs; (**b**) photos of commercially available melanometers (left to right) Mexameter MX 18, Skin colorimeter CL400, Dermaspectrometer DSM III, SkinColorCatch and CM700d spectrophotometer; (**c**) an illustration of the spectral reflectance methodology used to calculate melanin scores[17] (Copyright Wiley-VCH GmbH. Reproduced with permission); and basic colorimetry concepts including (**d**) CIE XYZ 2° and 10° standard observer color matching functions, (**e**) the CIELAB color space[11] (Reprinted with permission from Elsevier), and (**f**) a skin color volume on the L*b* plane (CIELAB 1976 system)[83] (Reprinted with permission from Dr. S. Del Bino).

The final type of melanometer discussed is the colorimeter, which quantifies skin color as perceived by the human vision system using the CIE standard observer model. The distributions that represent spectral sensitivity to light have been standardized as mathematical functions, namely 2° and 10° standard observers (Fig. 3 d). The 2° standard observer is typically used with colorimeters and the 10° standard observer is typically used with reflectance spectroscopic melanometers[11]. The 2° standard observer represents the average human eye's spectral sensitivity if viewing colors at an armlength distance from a small field of view whereas 10° standard observer represents visual assessment from a larger field of view and provides better correlation to human color vision[11]. Colorimeters typically deliver white light to the skin, then either apply RGB filters corresponding to known human eye spectral sensitivity or analyze spectral data from reflectance spectroscopy measurements[137,138]. Calibration of colorimeters is often performed using a white reference or a pair of black and white plates.

## Data processing methods and melanin metrics

As illustrated in Table 1, melanometers primarily generate two types of metrics—melanin/erythema indices and CIELAB colorimetry parameters. However, the underlying measurement methods and data processing

**Table 1 | Summary of melanometer characteristics**

| Device* | Device type | Wavelength (nm) | Key output metrics |
|---|---|---|---|
| CHECK3 (Datacolor) | Reflectance spectroscopy | 400–700 | L*a*b*, ITA [a] |
| CM600d (Konica Minolta) | Reflectance spectroscopy | 400–700 | Melanin conc. [a], MI[1][a], L*a*b* |
| CM700d (Konica Minolta) | Reflectance spectroscopy | 400–700 | Melanin conc. [a], MI[1][a], L*a*b* |
| CM-2002 (Konica Minolta) | Reflectance spectroscopy | 400–700 | Melanin conc. [a], L*a*b* |
| MA68II (X-Rite) | Reflectance spectroscopy | 400–700 | L*a*b* |
| USB2000+ (Ocean Insight) | Reflectance spectroscopy | 300–1100 | Melanin conc. [a] |
| S2000 (Ocean Insight) | Reflectance spectroscopy | 350–1100 | AUIC [2], 450-615nm [a] |
| CL400 (Courage-Khazaka) | Tristimulus colorimeter | 440–670 | L*a*b*, ITA |
| CL440 (Courage-Khazaka) | Tristimulus colorimeter | 440–670 | L*a*b*, ITA |
| CR-300 (Konica Minolta) | Tristimulus colorimeter | 450, 550, 600 | L*a*b* |
| CR-221-R (Konica Minolta) | Tristimulus colorimeter | 450, 560, 600 | L*a*b* |
| DSM III (Cortex Technology) | Tristimulus colorimeter / Narrowband reflectance | 460, 530, 620 / 568, 655 | L*a*b* <br> $MI^1 = \log_{10}\left(1/R_{655nm}\right) \times 100$ <br> $EI^3 = \log_{10}\left(R_{655nm}/R_{568nm}\right) \times 100$ [9,64,179] |
| DSM IV (Cortex Technology) | Tristimulus colorimeter / Narrowband reflectance | 460, 530, 620 | L*a*b*, MI[1], EI[3], ITA |
| Dermacatch (Colorix, N/A) | Tristimulus colorimeter / Narrowband reflectance | 460, 540, 620 | L*a*b*, ITA, MI[1], EI[3] |
| SkinColorCatch (Delfin Technologies) | Tristimulus colorimeter / Narrowband reflectance | 460, 540, 620 | L*a*b*, ITA, MI[1], EI[3] |
| Mexameter MX16 (Courage-Khazaka) | Narrowband reflectance | 568, 660, 880 | $MI^1 = 500/\log_{10}(5) \times \left[\log_{10}\left(R_{880nm}/R_{660nm}\right) + \log_{10}(5)\right]$ <br> $EI^3 = 500/\log_{10}(5) \times \left[\log_{10}\left(R_{660nm}/R_{568nm}\right) + \log_{10}(5)\right]$ [195] |
| Mexameter MX18 (Courage-Khazaka) | Narrowband reflectance | 568, 660, 870 | $MI^1 = \log_{10}\left(R_{870nm}/R_{660nm}\right) \times 1000$ <br> $EI^3 = \log_{10}\left(R_{660nm}/R_{568nm}\right) \times 1000$ [93] |

* A version of this table including the following device settings is provided as Supplementary Data 1: standard illuminants, standard observer, aperture size, and specular reflection inclusion (with illumination and viewing angles). The mention of commercial products, their sources, or their use in connection with material reported herein is not to be construed as either an actual or implied endorsement of such products by the Department of Health and Human Services.
[a] Custom derived metrics.
[1] Melanin index.
[2] Area-under-the-intensity curve.
[3] Erythema index.

algorithms and equations can substantially differ between melanometers reporting the same metric.

Narrowband reflectance melanometers typically generate melanin and erythema indices (MI,EI) using red to near-infrared reflectance measurements[9] following studies that showed estimated melanin content could be obtained from absorbance spectra A(λ) derived from reflectance R(λ) as A(λ) = -$\log_{10}$(1/R(λ))[82,95,139,140]. Since melanin absorption is the primary epidermal absorber in the 600–700 nm region (Fig. 3c), the slope of A(λ) in this region can be used to estimate epidermal melanin content[139]. Initially slope was based on reflectance near 650 nm and 700 nm[140]. Other studies have used reflectance at 630–700 nm[82,95], and 620–720 nm[12,17,141–143], and 650–700 nm[144]. Table 1 illustrates that commercial devices such as the Mexameter MX18 and DSM III use similar algorithms. It is also worth noting that one narrowband device (Skintel) used a different approach, based on simulations of light propagation in tissue[130]. While the names of metrics and wavelengths used in narrowband melanometers may appear similar, the lack of true standardization makes it difficult to perform effective inter-comparison of data collected by different devices.

Several variations on this approach to measuring melanin index have incorporated additional wavelengths. Area under the intensity curve along the 450–615 nm wavelength interval of reflected light has also been used to evaluate pigmentation and classify skin color[145,146]. Melanin density, derived by subtracting reflectance at 400 nm from 420 nm, is intended to eliminate the confounding effect of hemoglobin absorption which is similar at both wavelengths[147–150]. Single wavelength remittance at 390 nm (Fig. 2a) has been used to estimate epidermal melanin concentration given its shallow penetration depth and independence of blood oxygen saturation[151]. A method that uses a wide range of spectral reflectance data is proposed to be most accurate. Melanin and hemoglobin metrics have been determined from the skin absorption spectrum (~ 500–700 nm) via multiple regression analysis, where the absorbance spectrum is assumed to be a linear summation of the absorptions of melanin and hemoglobin according to Beer-Lambert law[152–154]. Diffuse reflectance models where skin is assumed to be a homogeneous semi-infinite turbid media have also been used to determine chromophore concentrations and light scattering properties of the skin[85,155].

Colorimetry devices employ reflectance spectroscopy or RGB reflectance approaches to classify skin according to its visual color appearance. Using the CIELAB color space (Fig. 3e)[156], reflectance measurements can be converted into colorimetric quantities. The CIELAB system involves a three-

dimensional color space consisting of three axes – L*, a* and b*, where L* represents lightness with values from 0 (black) to 100 (white), a* and b* represent the red/green and yellow/blue attributes on the chroma plane, respectively (Fig. 3e). Although applications of tristimulus colorimetric techniques include estimating visual appearance and chemical analysis[137], the CIELAB measurements have been found useful for quantifying skin color, with L* and b* representing pigmentation and a* representing erythema levels in a given individual[11,132,157–159]. Objective classification of skin color has also been attained by use of the Individual Typology Angle (ITA), which can be derived from L* and b* as follows[83,160–164]:

$$ITA° = \tan^{-1}\left(\frac{L*-50}{b*}\right) \times \frac{180}{\pi} \qquad (4)$$

ITA values have been utilized to categorize skin color. Early work on ITA[160] was limited to values above 10°, and these values were used to define four primary categories: very light, light, intermediate, and tan. Subsequent studies considered ITA values as low as –90° and added brown and dark categories (Fig. 3f)[163]. However, the ITA skin color bins for each category are non-uniform in width, with much wider bin size for darker skin compared to lighter skin. To develop ITA-based categories more closely corresponding to differences in epidermal melanin content, a strategy employing uniform

bin sizes may be more appropriate. Amongst all melanometry metrics, ITA is the most commonly evaluated, having been validated against many established approaches such as Fontana–Masson staining[161], HPLC[165,166] and spectrophotometry[165,166].

## Comparison between melanometry and subjective methods

Many melanometer studies have compared measurements to subjective classification methods such as FSP. While FSP is an imperfect technique, its prevalent use may enable comparison of data from different studies and thus facilitate standardization between different pigmentation evaluation methods or at least provide a basic check on melanometer validity. We compiled data on correlations between objective melanin metrics and FSP in seven devices (Fig. 4)[105,167–175]. Correlation coefficients between melanometer outputs and other subjective classification tools such as VLS scale and observer rated pigmentation scale have been evaluated for Mexameter MX18 (MI, R = 0.90)[176], Dermaspectrometer (MI, R = 0.32–0.63)[64] and Minolta Chromameter CR-221-R (L*, R = 0.23–0.51; b*, R = 0.24–0.48)[64]. Custom derived melanin metrics extracted from reflectance spectroscopic data have also been evaluated against FSP classification scale (R = 0.76 – 0.91)[61,145,146], visual subjective grading of pigmentation (R = 0.92)[95], and self-reported skin color (R = 0.113)[115]. Melanometers have shown high inter-

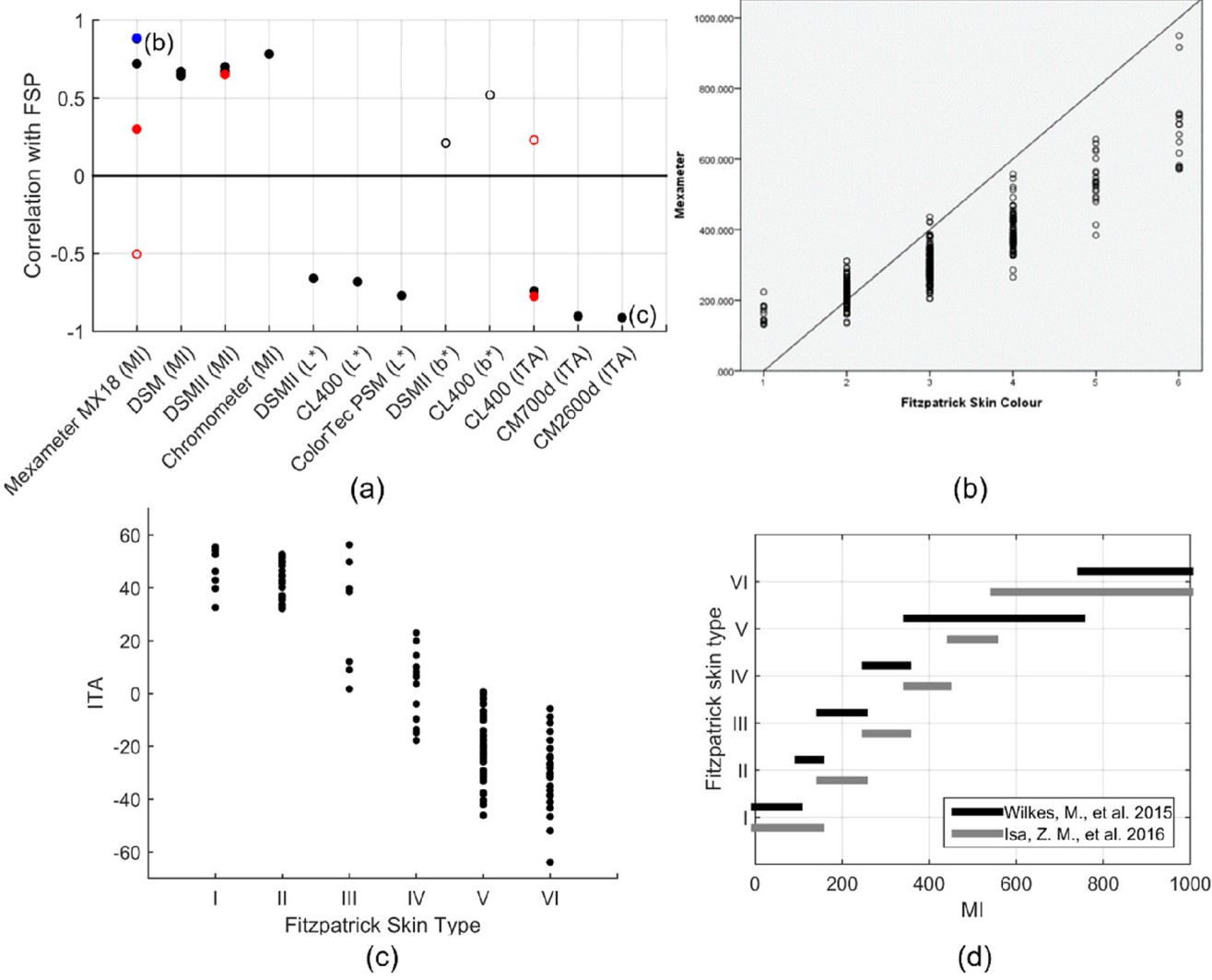

**Fig. 4 | Correlation between melanometer outputs and FSP scale. a** Correlation coefficients between FSP and several commercial devices[105,167–175,230] (filled – significant $p < 0.05$; unfilled—not significant $p > 0.05$). Marker colors represent different methods employed to determine FSP [black – questionnaire, blue—perceived FSP, red—perceived FSP plus questionnaire]; Correlation between FSP scale and (**b**) Mexameter MX18 MI[105] (**c**) CM2600d ITA[230]; (**d**) MI based FSP classification[105,178].

operator reliability[64,105] compared to subjective classification tools which are often affected by perceptual errors[105,170,174,177] and demonstrate poor agreement between observers[64].

Good correlation is generally seen between device outputs and FSP, although only approximately 28% of studies showed |R| > 0.75 (Fig. 4). Unlike MI, L* and ITA were inversely proportional to skin pigmentation levels, which is expected based on their definitions in section 4. b* was found to not show statistically significant correlation with FSP (Fig. 4a), which may be due to poor sensitivity to melanin in some pigmentation groups[159]. Although the moderate to good correlation results (Fig. 4a) indicate device agreement with FSP, the results were highly variable (|R| = 0.23–0.90).

Different methods have been employed to determine FSP (marker colors in Fig. 4a). FSP was determined using questionnaires in 8/11 studies (20/26 correlation results), using visual perception in 1/11 studies (1/26 correlation results), and using both visual perception and questionnaires in 2/11 studies (5/26 correlation results). While the lone study using visual perception of FSP alone reported strong correlation (R = 0.886, p < 0.001) between FSP and melanometer output[105], further comparative analysis between different FSP methods could not be performed due to limited studies. No significant variation in correlation results with respect to the heterogeneity of study population FSP types were identified in the compiled dataset. Some studies demonstrate an unconventional method of determining FSP/skin color from melanometer outputs[105,178] (Fig. 4d). Results illustrate moderate discrepancies in MI value ranges for each FSP level, with values from one group generally showing lower MI values for FSP II-IV.

## Comparison between melanometers

Given the wide variation in melanometer approaches discussed earlier, there is a need to investigate measurement agreement between different devices to understand device validity and soundness of the technology[13]. To this end, we compiled data on reported correlations between different melanometer outputs (Fig. 5a, b)[68,131,148,178–184].

Melanometry outputs were generally well correlated between different devices and metrics, with 76% (13/17 results) with |R| > 0.75 (Fig. 5a). Mexameter (MX16, MX18)[131,148,178,181,182,184], Dermaspectrometer (DSM)[68,179–181] and Minolta Chromameter (CR200, CR300)[179,181,183,185] were the most frequently correlated devices with other commercial melanometer outputs. Even metrics from different measurement techniques (i.e., colorimetry and narrowband reflectance) were well correlated (|R| = 0.56 – 0.98). However, a few studies have suggested that MI is a better predictor of melanin content than L*[68,179], particularly due to decreased correlation observed between MI and L* in more vascularized body sites of low pigmentation skin with high intra-individual variability (e.g., forehead, |R| = 0.93 vs inner arm, |R| = 0.98)[68]. This greater sensitivity to blood may be due to using shorter wavelengths—where blood absorption is high—to measure L*. Several custom melanin metrics derived from reflectance spectroscopy data have also been correlated against commercial melanometer outputs (|R| = 0.78–0.96)[60,61,148]. However, it should be noted that strong correlation between devices does not necessarily mean that either device actually measures the intended parameter[13].

## Repeatability and reliability of melanometry

Repeatability is a critical aspect of melanometer performance and may be affected by individual characteristics (age, sex, race, anatomical site, skin surface properties), intra- and inter-individual variability (temporal, physical and mental activity, orthostatic effect, menstrual cycle/menopause), environment conditions (lighting, temperature) and several instrument-related variables[9]. We compiled data for intra-observer[64,105,131,181,183,185–190], inter-observer[64,167,186,187,191] and inter-instrument[186] in vivo repeatability studies where intra-class correlation coefficient (ICC) and/or coefficient of variability (CV) have been reported (Fig. 5c–e). ICCs > 0.90 were considered to indicate excellent reliability, good between 0.75 and 0.90, moderate between 0.50 and 0.75, and poor <0.50[192], whereas CVs <10% were considered excellent, good between 10% and 20%, moderate between 20% and 30%, and poor > 30%.

Prior review studies have reported the Minolta Chromameter (CR200/CR300) to be a reliable device for skin color assessment due to its good intra-observer, inter-observer and inter-instrument repeatability compared to other devices[9,13]. However, these papers included a much smaller compilation dataset of melanometer reliability studies than presented here. The heterogeneity of skin pigmentation of the sampled population varies between different studies. In our compiled repeatability dataset (Fig. 5c–e), 5 out of 13 studies reported the FSP distribution of the study population, of which only 2 studies included participants of FSP VI. For the rest of studies, 5 studies reported race and 3 studies did not report any information on patient skin type or ethno-racial background. No significant trend was observed in the repeatability data for studies that included FSP VI versus studies that did not include FSP VI. As shown in Fig. 5c–e, excellent results (ICC > 0.90 and/or |CV| < 10%) were obtained in approximately 85%, 75%, and 67% of intra-observer, inter-observer, and inter-instrument studies, respectively. It should be noted that most reported repeatability studies tested intra- and inter-observer variations for devices and few inter-instrument repeatability studies have been performed. Only one melanometer measure (b* for DSM II) exhibited poor reliability values (intra-observer, |CV| = 63.9%; inter-observer, |CV| = 32.2%), probably due to the fact that the b* parameter is affected by both pigmentation (i.e., yellowness) as well as skin circulation (likely venous circulation, i.e., blueness) thus leading to greater variability[187]. However, b* has demonstrated moderate to excellent intra-observer, inter-observer, and inter-instrument reliability in several other studies (Fig. 5c–e). Considering all results, the repeatability of objective melanin metrics via melanometers were generally high, particularly when compared to subjective classification methods for pigmentation (ICC = 0.304-0.87, |CV| = 32.4–50.4%[64,167,187]).

## Cross contamination between pigmentation and erythema in melanometry

Commercial melanometers are intended to quantify melanin and erythema biomarkers, but they generally cannot perfectly isolate the contributions of tissue constituents[12]. Narrowband melanometers using red and NIR wavelengths are especially susceptible to these effects because their greater optical penetration in tissue increases sensitivity to hemoglobin located in deeper dermal layers (Fig. 2a)[12]. Due to spectral overlap of hemoglobin and melanin, measurements using a limited number of spectral bands may be subject to crosstalk artifacts[15,93,95,151]. Methods to separate melanin and hemoglobin signal contributions are typically necessary, yet imperfect. Although tristimulus colorimeter metrics (L*, b*, and ITA) are a measure of perceived skin pigmentation, CIELAB parameters were not intended to extract chromophore-specific information, and changes in melanin or hemoglobin concentration can impact all three indices measured (L* a* b*)[12].

The challenging nature of separating melanin and erythema content without interference is readily apparent when evaluating correlation between melanin and erythema metrics (see cross talk between melanin and erythema outputs for a few commercial devices in Supplementary Fig. 2). A high degree of correlation was shown in 85.7% (6 of 7) metric comparisons in commercial melanometers (|R| > 0.70; Table 2). These results indicate that melanometers may often produce higher erythema readings in darkly pigmented skin without any physiological rationale to support this finding. Studies of Mexameter system measurements during erythema induction by UV irradiation reported a significant decrease in measured pigmentation as cutaneous redness increased, despite actual skin pigmentation being, presumably, constant[131,182]. Changes in tissue blood volume during orthostasis or application of a pressure cuff have also been shown to affect objective melanin metrics[193,194]. As most commercial melanometers use contact-based probes to acquire skin reflectance, they include the possibility to exert some pressure on the measurement site which could affect the cutaneous blood content and hence influence the measured color[9,95]. A few instruments have introduced alternatives to reduce the effects of probe pressure which include incorporation of a plastic mask to distribute the force over a relatively large area (Minolta CM2002[95]), foot-switch model where measurements can be

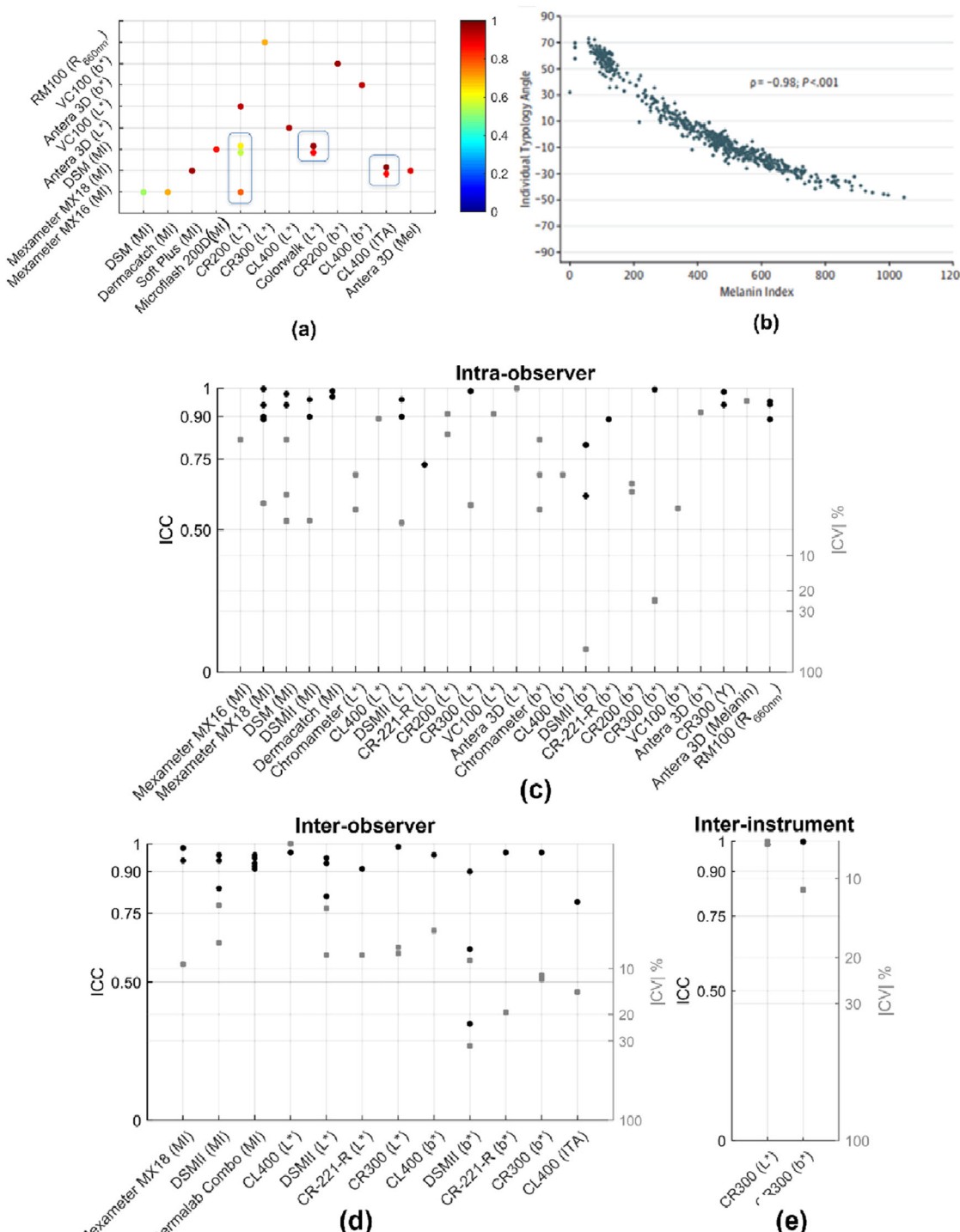

**Fig. 5 | Correlation between melanometers and in-vivo repeatability results for devices. a** Correlation coefficient between several pairs of commercial devices[68,131,148,178–184] and (**b**) correlation between ITA (CL400) and MI (MX18)[178]

(Reprinted with permission from American Medical Association); ICC and CV for (**c**) intra-observer[64,105,131,181,183,185–190], (**d**) inter-observer[64,167,186,187,191], and (**e**) inter-instrument[186] in vivo repeatability for several commercial devices.

performed without applying any pressure on the measurement site (Mexameter MX16[195]), and inclusion of elastic spring instrument in the probe that ensures constant pressure application on skin (Mexameter MX18[131,196], CL400[131]). Non-contact melanometers such as the SIAscope II and Antera 3D have overcome such limitations.

Diffuse reflectance spectroscopy (DRS) has been proposed to be a more accurate approach to measure skin chromophore concentrations because it offers rich spectral data and can be combined with light transport models and known chromophore absorption spectra to account for

contributions of multiple tissue absorbers[12]. DRS provides spectral data with a high information content, so it can estimate the concentration of biologically relevant chromophores (e.g., melanin, $HbO_2$, HHb) as well as relatively less abundant chromophores like bilirubin, methemoglobin, and carboxyhemoglobin[12]. However, DRS has still been reported to be sensitive to cross-contamination effects between melanin and erythema measures[60]. Individual studies have demonstrated fewer cross-contamination effects for SIAscope II[93] and Dermacatch[182]) melanometers. Single wavelength remittance at 390 nm, which predominantly probes the epidermis, is

**Table 2 | Crosstalk between melanometer melanin and erythema outputs for several devices**

| Melanometer | Melanin | Erythema | Correlation coefficient | Reference |
|---|---|---|---|---|
| Mexameter MX18 | MI | EI | 0.94 | 60 |
| Mexameter MX18 | MI | EI | 0.89 | 93 |
| Dermaspectrometer | MI | EI | −0.77[5] | 68 |
| Photovolt | L* | a* | −0.71[5] | |
| CM700d | MI | a* | 0.79 | 133 |
| | L* | EI | −0.71 | |
| | b* | EI | 0.09 | |

[5]Estimated from data graphically extracted from paper using WebPlotDigitizer software.

another approach that has been proposed to predict epidermal melanin concentration with less sensitivity to dermal blood volume[151].

## Comparison of melanometry to reference approaches

While many papers have compared melanometry measurements to subjective skin tone levels or other melanometer results, the most effective way to validate performance is in comparison to an objective, accurate, and well-established reference[187]. Validation against non-optical measurements involving direct assessment of tissue samples can provide a scientifically rigorous demonstration of the credibility of melanometers, while also correlating these metrics to biological entities having known physical properties. The primary disadvantage of reference techniques is that they are destructive and require invasive skin biopsy. Perhaps the most promising method for verifying melanometry is to benchmark measurements to melanosome volume fraction[16,84].

For decades, tissue melanin content has been measured using histological stain methods. The Fontana-Masson (FM) stain is widely considered the preferred histologic approach for identifying melanin and has been used extensively in both research applications and clinical histopathology (Fig. 6a)[161,197,198]. Alternative stains such as the Von Kossa (VK) and Warthin-Starry (WS) stains have also been used to quantify epidermal melanin content[93,198]. Although FM stain is popular, a recent comparative analysis of FM, VK, and WS stains has demonstrated WS stain to provide a more sensitive and specific detection of epidermal melanin compared to FM and VK stains[198]. Melanin content has been quantified from stained histological slides using different techniques such as mean score density/field using a continuous visual analog scale electronic meter[93], surface covered by the staining in the epidermis (Fig. 6a)[161,198], intensity of the black stain divided by the area of the epidermis[82], and proportion of total area of deep layers epidermis (corresponding to the first two cell layers above the epidermal-dermal junction) of skin sections stained as melanin[147]. While no standardized approach has been established to quantify epidermal melanin via histological methods and histological markers are subject to errors due to processing, interpretation, and sampling bias, currently they are the best available metric for melanin[187,198].

Non-microscopy approaches can also provide high quality results. High-performance liquid chromatography (HPLC) is perhaps the most well-established technique for quantifying melanin markers in tissue samples[199,200]. A highly sensitive and specific laboratory method, it enables melanin composition to be determined[81,165,166,200]. Eumelanin consists of 5,6-dihydroxyindole (DHI) and 5,6-dihydroxyindole-2-carboxylic acid (DHICA) moieties, while pheomelanin consists of benzothiazine (BT) and benzothiazole (BZ) moieties. These melanin monomer units can be quantitatively analyzed through specific degradation products by HPLC, including pyrrole-2,3,5-tricarboxylic acid (PTCA) and pyrrole-2,3-dicarboxylic acid (PDCA) for DHICA and DHI moieties of eumelanin, respectively. Pheomelanin moieties BZ and BT can be analyzed as their degradation products, thiazole-2,4,5-tricarboxylic acid (TTCA) and 4-amino-3-hydroxyphenylalanine (4-AHP), respectively. Figure 6b shows an

example HPLC chromatogram for dark epidermis. Quantifying moiety contributions from participants in six skin color groups based on ITA classification (Fig. 6c) indicated that proportions of PDCA (35%), PTCA (41%), TTCA (20%), and 4-AHP (4%) are constant regardless of skin pigmentation[166]. Melanin content can also be estimated spectrophotometrically using optical absorption methods at different wavelengths (e.g., 350, 409, 500, 650 nm) after solubilization of melanin from tissue samples in NaOH or Soluene-350[158,161,165,201].

Several studies have attempted to validate commercial melanometers against established measurements (Fig. 6g)[93,132,148,157–159,161,165,202]. Good correlation has been observed between ITA and epidermal melanin content measured via FM staining (R = 0.87, P < 0.0001, Fig. 6d)[161], HPLC (R = 0.927, P < 0.0001, Fig. 6e)[165] and spectrophotometry (R = 0.922, P < 0.0001, Fig. 6f)[165]. Correlation between total melanin content obtained by HPLC and spectrophotometric methods has also been assessed, with high correlation (R = 0.98, P < 0.0001[161]; R = 0.99, P < 0.0001[165]) observed between the two methods. Findings generally indicated good correlation, with 59% of the results showing |R| > 0.75. This is much better than for subjective skin color classification (28%), which supports the use of melanometers for a more accurate measurement of skin pigmentation. Among the common objective melanin metrics (MI, L* and ITA), 44% of L* and 77% of ITA results showed |R| > 0.75, with ITA being the most commonly evaluated metric. It should be noted that despite being a popular metric, MI has only been validated against reference measurements of melanin concentration (histological assessment) in one study, with a modest outcome (R = 0.33, P = 0.02[148]). It has also been observed that b* alone does not demonstrate a strong correlation with HPLC-determined eumelanin, pheomelanin and total melanin (Table 5 in[132]). Although melanin was reported as a major contributor to b* values in lighter skin types (R = 0.71, P < 0.00001), this relationship breaks down in darker skin types due to optical masking of yellow light by high concentrations of epidermal melanin[159].

A limited number of studies have validated custom melanin metrics from reflectance spectroscopy against established techniques[82,147,148]. Melanin content extracted from the linear part of the 630–700 nm absorbance spectrum demonstrated strong correlation (R = 0.80, P < 0.001[82]), whereas melanin density (R420nm - R400nm) demonstrated weak (R = -0.280, P = 0.05[148]) and moderate (R = 0.68, P < 0.01[147]) correlation when validated against FM-stained histological melanin content. Despite the benefits of comparing melanometers to established alternative methods, limited data is available, including for many commercial melanometers.

## Future directions

Advances in melanometer technology may facilitate efforts to ensure equitable performance in pulse oximeters and other optical devices. It may be possible to improve performance through incremental changes in established reflectance devices or greater changes using innovative emerging technologies. Here, we briefly discuss potential technologies that may address this need.

While we described several reflectance-based techniques above, this is only a subset of proposed methods or possible methods that could be developed in the future for quantifying skin pigmentation. Prior studies have measured reflectance at short visible wavelengths (e.g., 460 nm)[125] and calculated the difference between reflectance at 400 and 420 nm[147,151]. Given the high melanin absorption and small penetration depths offered by short visible wavelength reflectance approaches, it seems probable that such devices would enhance selectivity to the epidermis, thereby minimizing interference from dermal blood. Despite small penetration depths that should mostly avoid interrogating vasculature, hemoglobin absorption is high for short wavelengths (especially the Soret band from ~400 to 450 nm[91]) and thus may confound melanin measurements. This effect could be mitigated by using short wavelengths for which hemoglobin absorption is also minimized, e.g., 380–400 nm or 450–470 nm[151].

Photoacoustic Imaging (PAI) is a technology that exploits the rapid heating of blood vessels under pulsed illumination. PAI delivers

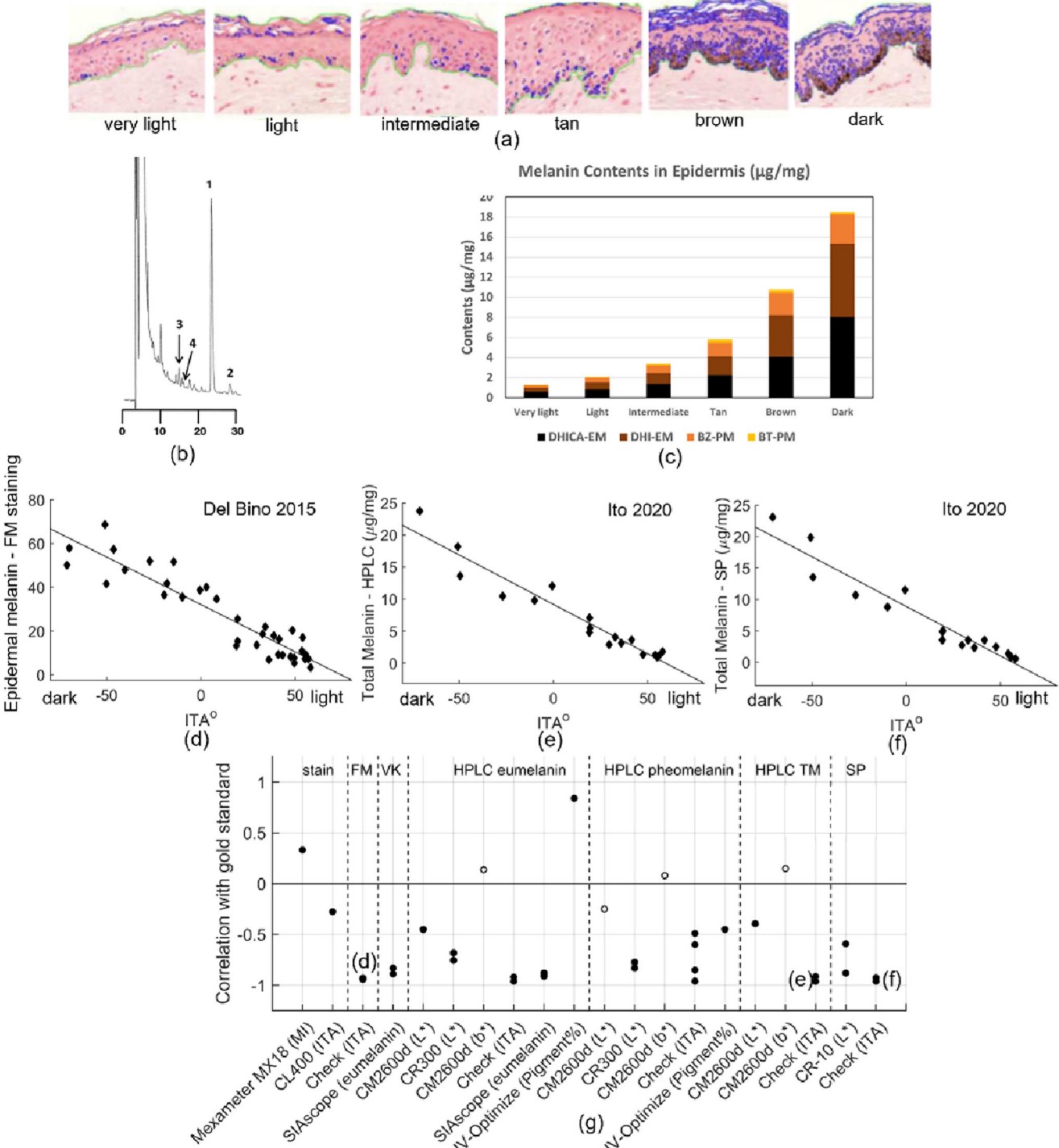

**Fig. 6 | Comparison of melanin measurement reference methods and correlation between ITA and reference melanin measures.** Comparison of methods that measure epidermal melanin content (**a**) an example of FM-stained skin biopsy sections shows increasing levels of melanin and non-specific staining in the stratum corneum[161] (Reprinted with permission from John Wiley and Sons) (**b**) an example HPLC chromatogram for dark human epidermis exhibiting melanin markers for DHICA, BZ, and DHI at peaks 1, 2 and 3, respectively[165], (**c**) contents of eumelanin (DHICA-EM, DHI-EM) and pheomelanin (BZ-PM, BT-PM) in epidermis in different skin color groups[166], correlations between ITA and reference melanin measures including (**d**) epidermal melanin by FM staining[161], (**e**) total melanin (TM) by HPLC[165], and (**f**) total melanin by spectrophotometry (SP)[165], (**g**) compiled correlation results between melanometer outputs and high-quality melanin content measures (filled—significant $p < 0.05$; unfilled—not significant $p > 0.05$)[93,132,148,157–159,161,165,202].

nanosecond-duration light pulses to tissue, where rapid absorption generates acoustic pressure waves detectable at the tissue surface by ultrasound transducers. Multispectral PAI can generate maps of tissue chromophore content, although quantitative PAI remains challenging due to complex spectral artifacts[203]. The potential for PAI to measure melanin concentration

in people of varying FSP has been established[204]. Strong correlation has also been shown between PAI epidermal signal intensity and ITA values measured with a colorimeter (SkinColorCatch) and FSP (although the methodology for determining subject FSP was not described)[205]. To optimize accuracy of melanin quantification, approaches based on photoacoustic

microscopy (PAM)[206] capable of sub-melanocyte-scale resolution may provide a more effective solution, and this approach has been investigated for quantifying melanin in ex vivo retinal tissue[207]. However, PAI and PAM systems are currently too complex and costly for widespread clinical adoption as melanometers.

Another promising approach for melanometry is spatial frequency domain imaging (SFDI)[208], which makes use of patterned surface illumination to perform multispectral, depth-selective reflectance imaging and map tissue absorption and scattering coefficients. Recently, SFDI measurements of epidermal absorption coefficients were well correlated with colorimetry measurements from a Chroma Meter CR-400 (Konica Minolta Sensing, Inc., Tokyo, Japan)[204]. However, absorption coefficients were underestimated for highly pigmented skin, due at least in part to the use of a homogeneous skin model in the algorithm as well as use of a calibration phantom that did not provide optically realistic representation of highly pigmented skin. Multi-layered models can help address some of these shortcomings as they can more accurately isolate layer-specific melanin concentration[209]. However, more advanced models are required to account for the inhomogeneous distribution of melanin in more heavily pigmented skin[89].

Several advanced high resolution optical imaging approaches might also serve as effective melanometers. Confocal microscopy can provide imaging of reflectance or fluorescence at micron-scale resolution and has been used to image melanosomes in the epidermis[210,211]. Multi-photon imaging to depths of several hundred microns can provide accurate estimation of melanin concentration in vivo, without the need for biopsy[209]. Electron paramagnetic resonance (EPR) spectrometry can detect melanin pigments present in skin, hair, and most types of malignant melanomas[212]. Near-infrared fluorescence has also demonstrated promise as an approach for quantifying epidermal melanin content[213]. Optical coherence tomography (OCT) represents the standard-of-care for clinical retinal imaging and is capable of achieving the resolution necessary for visualization of epidermal microstructure[214]. While restricted region of interest for characterization of melanin as well as cost and portability of advanced technologies may be a limiting factor to wide acceptance, progress in these technologies will likely lead to more practical options in the future. Alternately, they may be able to serve as high quality reference approaches to complement or replace histopathology as the recognized best practice for melanometer validation (i.e., gold standard).

## Best practices and standardization

Overall, the literature indicates that melanometry devices have the potential to be highly effective tools for pigmentation assessment. However, optimal results will require high quality devices as well as optimized methods for implementation and validation. As with any scientific instrument, it is essential that best practices be developed and widely applied in a consistent manner; thus, there is a critical need to incorporate these practices into international consensus standards.

Consensus guidelines for skin pigmentation measurement using colorimetry and reflectance spectroscopy devices have been published, however one of these documents was developed over 25 years ago (by the European Society of Contact Dermatitis, ESCD)[9] and two others focus on cosmetic sun protection testing[164,215] and provide similar content. These standards recommend that probes be applied with minimal force to reduce variations caused by displacing blood. This approach aligns with some prior studies[140], whereas others have applied firm[125] or moderate pressure[11]. Consensus recommendations also include triplicate measurements with the probe lifted from the skin after each acquisition, a maximum ITA standard deviation of 0.2, and the rapid execution of measurements with the probe steady and perpendicular to the skin. Direct sunlight should be avoided, and ambient temperature should be 19–23 °C. Additionally, measurements should avoid highly heterogeneous regions—such as those containing visible blood vessels, nevi, and scars—as well as regions with discolorations that may be endogenous or exogenous (e.g., tattoos), or those to which cosmetics or medications were applied. Recommended frequency of

recalibration—ideally, a process described by the device manufacturer—has varied from before every use[11], after every 30 subjects[125] or after battery replacement[9]. In terms of requirements for the melanometer itself, ISO standards[164] recommend use of systems with a field of view of at least 8 mm and a lamp with color temperature of 6500 K. CIE colorimetry standards developed for non-biological use also provide recommendations on device specifications[156].

Establishing best practices for evaluating skin pigmentation during clinical studies is critical to enable reliable assessment of disparities. The University of California San Francisco (UCSF) Open Oximetry project includes a clinical trial evaluating pulse oximeter performance in a balanced diverse population where skin pigmentation is quantified using multiple color scales as well as colorimeters[216]. To directly address the optical effects of melanin on pulse oximetry, melanometry should be performed at tissue sites where the oximetry sensor is applied. In prior studies, data have often been collected at sites where melanin concentration is potentially high, such as the arm[60,125] head[217] or torso[218]. However, pulse oximeters are commonly used on regions of the index finger (palmar surface and nail) where the range of pigmentation levels is relatively small[204]. Studies with the Mexameter[196] and Chroma Meter CR-400[204] provided quantitative evidence that the palmar hand and finger have lower pigmentation than the ventral arm (e.g., melanin index of 42 vs. 240; L* of 60 vs. 39). Furthermore, values measured in the palm of otherwise highly pigmented people were only slightly different than those from people with low pigmentation. Melanocyte content in the nail bed has been measured as 5% that of normal skin and these cells typically do not produce melanin[219]. The ability to reliably differentiate between low levels of pigmentation in these distal finger sites may then be critical to evaluating impact. While the precision of most melanometers may be sufficient to measure skin sites exhibiting a large melanin content range, including skin proximal to the nail bed, accurately measuring low-pigmentation sites on the finger probably requires greater precision. Data acquired from a commercial spectrometer system may be capable of accomplishing this task effectively [personal communication with Dr. L. Shmuylovich; March 18, 2024]. It is not currently clear whether measurements at higher melanin content sites (e.g., dorsal arm) could serve as a viable surrogate for finger sites. Another practical issue for measuring finger sites includes the challenge of measuring strongly curved surfaces or small regions with bulky instruments and devices that have a large field of view.

It is worth noting that this discussion of measurement site is largely predicated on a racial disparity mechanism involving optical absorption by melanin. However, Monk et al. indicate that perceived colorism, or human perception of skin color by oneself and/or others[220] is linked to disparities in health (e.g., blood pressure) that may in turn impact device accuracy[221]. Other researchers have noted that skin color is a phenotypical trait that may be associated with other traits having physiological implications such as variations in vascular response[222] or hemoglobinopathy[223]. To directly test these mechanisms as sources of device performance disparity, measurement sites such as the face may be more appropriate, possibly in combination with racial category.

Melanometer calibration and validation represent another critical topic covered by the ESCD report[9]. Recommended calibration targets are typically very high and low diffuse reflection, i.e. white and black targets. Numerous articles have noted the use of such targets, particularly with commercial melanometers[93,131,181]. Fullerton et al. describe the use of white, pink and red plates for repeatability testing[9]. The ESCD report briefly mentions a nine-color calibration plate for colorimeter variability testing. Other studies have implemented sets of colored tiles (e.g., X-Rite Color Checker with 14 tiles)[181,182,224] for performance validation. When using color tiles for colorimeter testing, a color difference approach:

$$\Delta E_{ab}^* = ((\Delta L^*)^2 + (\Delta a^*)^2 + (\Delta b^*)^2)^{0.5} \qquad (5)$$

where $\Delta$ denotes the difference between reference and test devices) is recommended[156]. The ESCD report also recommends testing repeatability by measuring a white calibration plate 30 times in 10 second intervals to

ensure that the standard deviation of the color difference (ΔE*) is less than 0.07[225]. Although targets are commonly used for colorimeter assessment, they are probably insufficient for rigorous validation of colorimeters or spectroscopy devices intended for skin measurements. Research has indicated that color tile evaluations overestimate colorimeter accuracy in skin[181,182]. Additionally, color targets are essentially homogeneous, surface reflecting targets, whereas skin is a multi-layer turbid media into which light may penetrate up to a depth of centimeters. Even colorimeters that show strong inter-device agreement using targets might produce different ITA values in skin if they have different illumination geometries (wide-field imaging vs. point measurement), due to differences in sensitivity to tissue layers. Thus, the use of standardized multi-layered tissue phantoms[143] could facilitate standardization of diverse systems. Such phantoms would likely have to incorporate epidermis-simulating layers with a wide range of pigmentation levels[21].

Colorimeters and reflectance spectroscopic melanometers can be operated in different settings. CIE has reported recommendations on the use of standard illuminants, standard colorimetric observers as well as illuminating and viewing conditions for basic colorimetry [226]. It should be noted that device settings such as illuminants, standard observer, specular component inclusion/exclusion, and measurement geometry can impact the device output and the estimated melanin content. Therefore, ideally the same device settings should be used to compare values obtained with different colorimetric and spectrophotometric instruments[11].

Ideally, melanometers would report a standard metric with direct optical and biological relevance. Since ITA is based on the CIELAB color space it is often considered is a standardized metric, yet prior validation of colorimeters has relied on color charts rather than realistic turbid media. Additionally, ITA and MI do not have a direct biological meaning. A more broadly useful metric may be a form of the melanosome volume fraction ($M_f$) parameter used by Jacques[17] to calculate epidermis optical properties. Since melanometers do not measure epidermal thickness or provide a direct measurement of $M_f$ it may be useful to establish an effective melanosome volume fraction parameter, $M_{fe}$, based on an assumed epidermal thickness (e.g., 100 μm). By calibrating ITA or MI outputs to $M_{fe}$, direct comparisons between different devices would be possible. A similar calibration of photothermal sensing systems based on histology has been proposed to enable estimation of epidermal melanin mass per volume[151].

## Concluding remarks

Considerable published evidence supports the use of optical melanometers for assessing skin pigmentation level. Melanometry enables more accurate evaluation of skin pigmentation level than subjective approaches, with a high degree of repeatability and inter-device consistency. Subjective approaches may be sufficient in some clinical studies of optical device robustness to skin pigmentation, yet melanometry will likely enable detection of disparities with greater reliability or with smaller population sizes. Additionally, limited studies indicate strong correlation to objective established measures of epidermal melanin content, which may facilitate more rigorous scientific understanding of clinical results. While devices based on colorimetry and multi-wavelength spectroscopy provide relatively simple and practical tools, higher performance spectroscopy systems appear to enable the greatest accuracy and flexibility. The latter may also be particularly useful for pulse oximetry studies that address the low pigmentation levels in the distal finger.

Moving forward, we see a need for two main research directions. The first is to standardize melanometry through the establishment of best practices for human subject measurements, as well as benchtop and clinical validation of device performance. This should include establishment of approaches to characterize robustness to confounders (e.g., hemoglobin) shown to degrade accuracy. The second is to advance the accuracy of melanometry through improved instrument design and algorithms for reflectance-based devices, and development of alternate optical sensing technologies. Despite these challenges, optical melanometry has the potential to provide effective tools suitable for evaluating robustness of pulse

oximeters and other optical diagnostic devices to skin pigmentation, ensuring medical devices achieve healthcare equity in all patients.

## Data availability
The data that support the findings in this study are derived from published, peer-reviewed manuscripts. The source data underlying Fig. 4a, 5a, c–e and 6g is provided in Supplementary Data 2 to 5. All other relevant data are available from the authors upon request.

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

## Acknowledgements

We gratefully acknowledge support for this work from FDA's Office of Minority Health and Health Equity. We are appreciative of many useful discussions with the IEC/ISO Joint Working Group on Oximetry, Mr. Bob Kopotic (Edwards Lifesciences), Dr. Wim Verkruysse (Philips Research), Mr. Paul Batchelder (Clinimark), Dr. Leo Shmuylovich (Washington Univ. School of Medicine in St. Louis), Prof. Tony Durkin (Beckman Laser Institute, Univ. of California, Irvine), Prof. Yu Chen (Univ. of Massachusetts, Amherst), and Prof. Kung-Bin Sung (Taiwan Univ.). We would also like to acknowledge Dr. Wei-Chung Cheng's (FDA) expertise in color assessment and contributions to the paper.

## Author contributions

S.V, W.C.V., S.W., and T.J.P. completed the review, extraction, and quality assessment of papers. S.V. and T.J.P. drafted the initial version of the manuscript. W.C.V. and S.W. provided critical intellectual feedback. All authors approved the final version for publication.

## Competing interests

The authors declare no competing interests.
