## [Peer review file · Communications Medicine]

Melanometry for objective evaluation of skin pigmentation in pulse oximetry studiesReviewers' comments:

Reviewer #1 (Remarks to the Author):

I congratulate the authors on a very thorough review of the literature for a very important and timely topic. Overall I think this review is well written and well structured. If it could be shorted a little bit I think that would help for the general reader. I do have specific comments throughout consisting of some technical points that are worth refining in my view. I also at times found that the references used to support certain statements were not accurate and would recommend the authors consider rephrasing to be more accurate.

Introduction

- Occult hypoxemia is defined as 'undetected low SaO₂', but to the general reader it may be better to clearly spell out that that SpO₂ is normoxemic in the setting of SaO₂ hypoxia.
- Line 38 – opportunity to include recent important reference (Matthew D. Keller & Brandon Harrison-Smith & Chetan Patil & Mohammed Shahriar Arefin, 2022. "Skin colour affects the accuracy of medical oxygen sensors," Nature, Nature, vol. 610(7932), pages 449-451, October.)

- "These publications assert that such an approach will better capture skin pigmentation as a continuous variable, rather than lumping skin tones into discrete subjective categories (e.g., light, intermediate, dark), thereby facilitating enrollment of study subjects that adequately represent the full range of skin pigmentation levels." This statement includes publication [50], which is the APSF statement. I can't see any call for capturing skin pigmentation in a continuous fashion in that statement. If anything that statement is suggesting that there are other variables beyond pigmentation known to affect pulse ox accuracy, and that the 15% inclusion criteria should be reconsidered in light of the Sjoding paper.

Section 1 - Skin Optics

Figure 1: This diagram has issues with respect to accuracy, as it shows the stratum corneum and epidermis as having the same size? In reality the epidermis is about 5 times thicker than the stratum corneum. In Figure 2 the relative size of stratum corneum to epidermis is much more reasonable. Please replace Figure 1 skin layer schematic with something more accurate, you can check dermatology textbooks or dermatopathology textbooks.

In the diagram the depth of penetration of light has MIR called <1500 nm. That is incorrect. While some sources will extend NIR above 1000, recent work defines short wave infrared (SWIR) to 1700nm or potentially extending to 2.5 μm, and MWIR is typically 3-5 μm (with LWIR above 8μm). It's true that around 1400nm there is a decrease in penetration depth due to water absorption peak, but penetration depth at 1300nm exceed that at 1000 nm, and that is counter to what the diagram shows. Recent work confirms this point <https://www.ncbi.nlm.nih.gov/pmc/articles/PMC5147011/>

"Supranuclear caps form in response to UV" -

<https://www.sciencedirect.com/science/article/pii/S0022202X15400855?via=ihub>. I think when the authors say "in highly pigmented skin, melanin often forms..." readers may think that this is only in people with highly pigmented skin. Please clarify this point.

The impact of pigmentation being equivalent to SPF 13 is reported in the literature, but I'm not sure where the 70% reduction in skin cancer number comes from. Skin cancer is very rare in pigmented patients and there is no evidence to show that additional SPF protection is needed for skin cancer prevention in pigmented patients. The skin cancers that affect patient with pigmented skin occur often in areas that are not sun exposed and sadly are caught in later stages (and have higher mortality). The point about SPF and skin cancer I think is not needed for this review and should be cut.

I think that the typical sizes of melanosomes should be included rather than just saying that the melanosomes are larger for pigmented skin, this is especially relevant given the potential increased scattering coefficient observed in pigmented skin.

Figure 2:

Figure 2a quality is low – the black background with hard to read white text is not ideal. Bologna Textbook of Dermatology Edition 4 Chapter 65 Figure 65.5 has a much clearer picture illustrating the point. I have included it as an attachment for reference.

Figure 3: to those not experienced looking at the skin histology, it may have to say “basket weave” to describe the stratum corneum (i.e. “..seen as thin basket weave superficial layers”)

The authors suggest that the scattering effects of melanin are too small to impact optical measurements. I encourage them to add this study to their review <https://pubmed.ncbi.nlm.nih.gov/19687939/> where in vivo measurements were made to assess μ_a and μ_s' in subjects with varying pigmentation. It appears that there is a difference in μ_s' between light and dark subjects that is more pronounced at red than infrared wavelengths. This suggests that scattering could play a role. This would be consistent with the discussion by Chetan Patil in this work Matthew D. Keller & Brandon Harrison-Smith & Chetan Patil & Mohammed Shahriar Arefin, 2022. "Skin colour affects the accuracy of medical oxygen sensors," Nature, Nature, vol. 610(7932), pages 449-451, October.) In that op ed and in his work presented at SPIE Chetan showed that it is the inclusion of scattering properties of melanin with Monte Carlo models that is necessary to recapitulate the Sjoding results in silico. To my reading the authors come off as dismissive of scattering and focus completely on absorption. This implies that the bias is simply a signal to noise problem and not a problem of differential pathlength due to different scattering between red and IR channels for light and darkly pigmented skin.

For the description of FPS scale, I think it is useful to know that it was designed for the purposes of predicting what UV dose could be given safely to white patients undergoing systemic treatment with psoralen for their psoriasis. It was necessary because estimating a safe dose was deemed not possible to be done by just looking at someone's apparent pigmentation, and instead required a survey regarding the propensity to tan and/or burn. Figure 6a is an excellent figure to include, but I think the purpose of that figure in the perspective that Fitzpatrick wrote is to show that a Fitzpatrick skin type is defined by the change in skin with sun exposure – Type 1-4 are indistinguishable from facultative pigmentation alone and can only be differentiated when considering the skin tone after sun exposure. The statement “An FST skin type can correspond to different skin colors depending on whether constitutive or facultative pigmentation is present” implies that there is utility to defining an FST visual scale based on constitutive pigmentation, but the Figure clearly demonstrates that there is no difference (and therefore no information to be gleaned) for facultative FST 1-4.

I would strengthen this statement “While skin color charts based on the FSP method have been used [104, 105], these represent a non-standard implementation that contradicts the original intent of FSP to assess differences in sensitivity to sunburn for individuals with similar skin color”. One challenge in particular is that there is no established color palette for the FSP I-VI categories. These categories have not been mapped to a color space, and their standardization for what colors are used by various studies that report enrolling subjects based on Fitzpatrick type.

In general FSP is used ubiquitously as a visual scale despite the fact that it was specifically designed to measure something that was not assessable for dermatologists by eye. To make clear this problem, I suggest that the authors call any study that uses FSP from visual estimation a ‘perceived FSP’ rather than an FSP that is defined as originally defined through a set of questions about skin reactivity.

I know this may be a controversial point and one that the authors want to avoid bringing up, but the Von Luschan tiles are on display in the Holocaust museum as they were used by the Nazi's to characterize skin color in racial studies <https://collections.ushmm.org/search/catalog/irn564926> and Von Luschan was also a member of the German Society for Racial Hygiene. At the same time he appears to have opposed scientific racism in his work <https://www.jstor.org/stable/41157700>. I don't know if it's possible to bring all this up and it's not relevant to the review, however not mentioning the complicated past of Von Luschan's tiles may serve to alienate some readers. Is there something the authors can say to put this in appropriate context?

Line 221-224 are confusing: "Subjective methods such as FSP use a relatively coarse categorization for a continuum of skin tones (fair to dark) [120] which causes individual categories such as FSP VI to cover a wide range of melanin concentrations (and reflectance values, as shown in Figure 5)". I think the authors are making the point that a visual perceived FST fails to capture details regarding how skin pigmentation changes in response to sun exposure. However if FST is measured appropriately with questions about skin reactivity and not by color matching then information about how skin changes in response to sun exposure is captured. The authors reference this paper <https://jamanetwork.com/journals/jamadermatology/fullarticle/1737180> and in Table 3 there is a nice summary of proposed questions to improve FST, but these questions are not visual color matching based and are in my view in the spirit of what FST was designed to do.

I appreciate the authors include the work of Pichon et al (ref 125), but I think a key message of that paper was that the questions about sun burning and tanning that were designed for white people were difficult to interpret and not relevant to people with type V or VI skin. So in that paper it wasn't physicians being unreliable in assigning FSP, but it was that the questions themselves made self-assignment by subjects unreliable. This is an important point about the inequities that can arise in medical tools when the tools are not built to serve a diverse population, and I think is in line with the points that underpin the need for this review in the first place.

Section 4:

Table 1. In this table the XRite spectrophotometer is not listed, despite it being the spectrophotometer of choice in Figure 5. I know not every device can be listed but given it's presence in Figure 5 it seems appropriate to include. Also the CM700d is listed as having spectral range from 300-700. I think this is an error, the range is 400-700 (<https://sensing.konicaminolta.us/us/products/cm-700d-spectrophotometer/>)

In the text the authors state that examples of tristimulus colorimeters can be found in Table 1, but in Table 1 they also describe devices as having 'tristimulus colorimetry' applied during processing. I think using 'tristimulus colorimetry' to describe the processing step that occurs within the device is potentially confusing to non-expert readers. In Reference 66 from Ly et al they take a different approach – they call devices like the Delfin a tristimulus colorimeter. In numerous publications and device descriptions I've seen I also see the term colorimeter to mean a device that is designed to measure the color that is perceived by the human eye. Ly et al basically call a tristimulus colorimeter a type of device that measures spectral response across 3 narrow bands corresponding to the bands that the human eye perceives.

It is true that a spectrophotometer uses principles of tristimulus colorimetry to extract tristimulus values from a more complete reflectance spectrum. However I like the approach of Ly et al to make clear that the wide band spectrophotometer contains more rich information than a narrow band spectrophotometer or tristimulus colorimeter, and that tristimulus values can be mathematically extracted from the full spectrum. In fact for a broadband spectrophotometer one can apply the color matching equations for an arbitrary illuminant and observer angle, deriving different tristimulus values for each given illuminant and observer. You can not do that with a tristimulus colorimeter which typically only has one illuminant. For this reason I think the term tristimulus colorimeter should be

reserved in Table 1 to the RGB spectral reflectance devices like the Delfin.

One thing that is missing from Table 1 is a mention of whether devices include or exclude specular reflection (some allow one to choose, others give no choice), and also what the aperture and illuminant is for various devices. The CM700d has different aperture sizes, given that there is interest in determining a standard approach for melanometry, I suggest the authors include information comparing differences between device apertures or source-detector distances. These are variables that could affect the device output and therefore could have an impact on estimates of melanin content.

Figure 8 – It is good to show the different instruments, but a scale bar or at least showing them all in the same view would be helpful. The device in Figure 8e is much more compact than the device in 8d but you wouldn't know by looking at the figure.

Figure 10 – I could not find the color matching functions in 10a within reference 66 or 162. I think a better graph for that comes from Konica Minolta's website

<https://www.konicaminolta.com/instruments/knowledge/light/concepts/07.html#:~:text=4.1>

TRISTIMULUS COLORIMETRY, of these three primary colours. Or this is a resource that the authors may find interesting to share or review – it has some example code for implementing the conversion of a spectrum to an XYZ tristimulus value set and also shows the color matching functions.

<https://www.fourmilab.ch/documents/specrend/>

Figure 10c is reprinted from multiple papers, but I think the equation in the bottom is written in a confusing fashion : $ITA^\circ = (\text{ATAN}(L^*-50)/b^*)$... This way of writing the equation does not make clear that you are taking the ATAN of the ratio (i.e. $\text{ATAN}[(L^*-50)/b^*]$). While the authors do provide the correct equation in the manuscript, I think including the figure with this confusingly written equation could perpetuate errors.

The authors make a good point about ITA bins. "However, it should be noted that the ITA skin color bins are non-uniform in width with much wider bin size for darker skin compared to lighter skin." However, I think the caution should be expanded, because just as was the case with FST, type I-IV ITA span the range for Caucasian people, and V and VI are left for all people with pigmented skin. These bins have been used in multiple studies but I think it is important to clarify their history just as they clarified the history of the FST bins. The initial definition by Chardon in 1991 had 4 bins, 10° as the smallest bin cutoff and only included Caucasian subjects with skin varying from very light to tan, and $<10^\circ$ was considered brown. Only about 20 years later did the Del Bino work introduce the -30° cutoff for 'dark'. Just as darkly pigmented skin was an afterthought for FST, so too was it for the ITA bins.

Section 5:

One potential weakness of prior studies that compared FST to melanometer-derived values is that the application of FST is inconsistent – sometimes it is a perceived FST based on color matching to a non-standardized FST color palette, and sometimes it is based on asking questions about skin response to UV. In Figure 11 it would be good to know whether the studies listed used perceived FSP or questionnaire defined FSP – do the cases where correlation is strongest represent cases where it was perceived FST? Indeed the "observer rated pigmentation scales" had an impressive correlation coefficient of $R=0.92$, so perhaps FST agrees better with melanometry when FST is used as a visual scale.

I think the recent paper by Osto et al is quite excellent as a reference

<https://onlinelibrary.wiley.com/doi/abs/10.1111/php.13562>. In this paper FST was determined by dermatologists in a photomedicine unit, meaning that it was used as originally intended, and ITA was measured by spectrophotometer. The figure from this paper is better than the figure in 11c which doesn't include Type 1 and comes from a paper whose primary goal was not to determine how FST

and ITA relate. Also the paper from 11c does not describe how FST was determined.

Minor points: Line 360, 364, 366 Figure 11 is listed as Figure 111. Line 373 lists Figure 12 as Figure 122.

Figure 12: There are multiple errors in Figure 12b. First of all, the categories are reversed. The 'very light' category is put in the $<-30^\circ$ bin but should be in the $>55^\circ$ bin. The order of the y-labels must have gotten switched accidentally. Also the Chardon 1991 paper had no -30° bin, that was only 20 years later by Del Bino. I'm not sure what the purpose is of Figure 12b though, it just shows a visual representation of the ITA bins which were already described in the manuscript early. Is the goal to show that the ITA bins mirror the MI bins in the Wilkes study? That may be the authors listed the ITA bins light to dark to match the FST 1-6, but this creates a problem because ITA decreases as MI increases, so you won't be able to easily make the graphs look the same.

Section 7:

Good job summarizing lots of data and studies. One thing that is not clear from this data though is whether the results apply to subjects with dark pigmentation. Given that these subjects are often not well represented in studies, I think it is useful to know if these subjects are represented in these melanometer validation studies. The DSMII study for example (Ref 182) has 82% Caucasian subjects with 0 subjects with type VI FST. I am not sure about the other studies included in Figure 14, but I think it's important for the authors to look into that to provide some sense of how applicable these validation studies are for skin of color patients.

Section 8:

Line 457 – missing reference

Section 9:

Few typos (Line 509 Figure is called Figure 122b, Lin 529 Figure is called Figure 177, and many other examples where the 1s digit is repeated in calling a figure), but otherwise this was an excellent section and one of the strongest parts.

Section 10:

The authors do an excellent job finding relevant literature, but could improve the text by being more accurate. Reference 201 is described as "reflectance spectrum mean slope over 620 to 720 nm". That is incorrect however, because they say: "By fitting the logarithm of the ratio of the remittance spectrum from vitiligo involved skin to normal skin of the same volunteer, from adjacent sites, we obtained a straight line fit to the experimental data". So in reality they took the log of the reflectance spectrum and found the slope of that over 620 to 720. In essence this is a ratio of reflectance at 720 to reflectance at 620.

And in reference 202 for example the authors described it as a 'difference between reflectance at 650 and 710'. This is not accurate either, in that work they define melanin index as follows "Specifically, the melanin index is defined by the slope of OD (λ) in the region above 620-640 nm (Fig.2): $M=100(OD_{650} - OD_{700})$ ". It is a difference of the optical density, which is the log of reflectance, so it is in reality a ratio of reflectance at 700 and 650 since a difference of a log is equal to the log of the ratio. The work by Dawson (ref 138) is also a difference of the log of reflectance at 650 and 700, so it is identical to the approach described in 202 and is also mentioned in ref 201.

Indeed all of these approaches are just a slope of the linear absorbance spectrum (the log of reflectance) above 650, which is one of the methods the authors already described. So while the authors describe this work as difference from section 4, it is identical to what was previously described. This is an easy mistake to make (missing that a logarithm is applied), but critical that the

authors go back and ensure that alternative methods described in section 10 are indeed different.

Section 11:

One of the issues that the authors do not address is the practical aspect of measuring the nail bed. Many devices are so bulky that it is hard to position reliably at the nail bed, or the devices have measurement windows that are larger than the size of a nailbed. This can lead to an otherwise reliable device being very unreliable due to challenges with the site of measurement.

"However, since pulse oximeters are commonly used on the index finger, the range of pigmentation levels can be expected be relatively low [210]" The authors here mean the palmar side of the finger right? The 210 reference used SFDI and measured at the palm and forearm – there were no measurements done at the dorsal finger. The dorsal finger is a sun exposed site so should have similar pigmentation to forehead. Is it the case that the pulse ox signal only goes through the nail bed and doesn't go through the finger beyond the eponychium? It is hard to know exactly where a pulse oximeter is positioned on a patient, but this is an x-ray that may be instructive. <https://radiopaedia.org/cases/pulse-oximeter-1?lang=us>. The sensor position may be between the DPI and the tip of the distal phalangeal bone, wouldn't that be close to the eponychium and therefore include skin beyond the nail bed?

In the conclusion too the authors suggested that the range of pigmentation is small at the measurement site and I'm not sure we know this because the dorsal finger likely is part of the pulse ox measurement site.

Reviewer #2 (Remarks to the Author):

To clarify the potential for disparities in pulse oximeter accuracy used in healthcare and due to evidence of racial disparities especially in patients with dark skin, the manuscript addresses the need for objective methods to quantify melanin content in skin. After an introduction on the optics of skin pigmentation, the authors review the literature on melanometry and commercially available devices used as tools for objective skin pigmentation assessment. They compare the melanometry approach to subjective skin color classification methods i.e. self-reported skin color, race/ethnicity or other subjective methods. They also compare melanometry to gold standard melanin quantification methods i.e. Fontana-Masson staining, HPLC, spectrophotometry. The authors also discuss further directions that could improve these reflectance-based techniques and finally give the best practices and standardization that could enhance the rigor and quality of clinical studies on pulse oximetry and other optical technologies.

Overall impression of the manuscript:

The manuscript is interesting, well written, pleasant to read. It provides quantitative analysis data from the literature to assess the effectiveness of melanometer approach, giving the advantages and drawbacks of different commercially available devices and their usage validity in comparison to other objective quantification methods as well as subjective ones. The article identifies the best practices for standardized melanometer implementation and gives future directions to optimize the accuracy of melanin quantification.

This work really brings something to clinicians and researchers, not only for pulse oximetry studies. Even if racial bias in the latter is the starting point for this work, I wonder if "Melanometry for Objective Evaluation of Skin Pigmentation" wouldn't be a more appropriate title for the manuscript.

Comments for transmission to authors:

Comment 1: Line 123 "(...) whereas Asian skin contains approx. 2-fold higher melanin than Caucasian skin". Authors should qualify these statements and stress that classifications based only on race/ethnic origin and not on skin pigmentation could be misleading. The reader should not take these data for granted. Skin from donors of Caucasian and Asian descent with the same phototype or ITA° have the same (eu)melanin content. Furthermore, Asian skin comprises skin of individuals from Northeast Asia (Japan, Korea, China...) and Southeast Asia (India, south of China, Indonesia, Malesia...). So the 2-fold higher melanin content might be true when comparisons are done between European skin and Indian skin for example but not when comparing European to Japanese, North Chinese skin etc. Furthermore, in each country there is a variability in skin tones as described in Figure 2 of Del Bino and Bernerd British Journal of Dermatology (2013) 169 (Suppl. 3), pp33–40. So when it comes to Asian skin it should be specified from which geographic area and of which color.

Comment 2: Table 1 The last line on CHECK3 comes after the legend: this should be inverted.

Comment 3: Figure 6 is blurred, can you increase the quality?

Comment 4: Line 397 "However, a few studies suggested the MI appears to be a better indicator of melanin compared to L*, particularly due to decreased correlation observed between MI and L* in more vascularized body sites with high intra-individual variability" : "more vascularized" or "darker skin tones"? L* represents the level of grey the lightness of the skin while b* represent the level of yellow or blue of the skin. For objective skin color assessment with all the devices described it should be recommended to avoid measuring uneven skin areas i.e. blood vessels, nevi, stretch marks etc. Furthermore "MI appears to be a better indicator of melanin compared to L*" : this is logical because melanin content is different from skin color and L*b* and ITA ° measures skin color and not melanin content.

Comment 5: Line 529 "Figure 177 and b" should be Figure 17

Comment 6: Line 529 "Figure 17 a and b illustrate a typical HPLC chromatogram for dark human skin and the contribution of eumelanin (PTCA, PDCA) and pheomelanin (TTCA, TDCA) (...)". The latter should be (TTCA, 4-AHP) not (TTCA, TDCA) as illustrated in Figure 3a of Del Bino et al Pigment Cell Melanoma Res. 2022;00:1–5 that shows the contribution of both types of eumelanins : DHI-CA eumelanin and DHI-eumelanin, assessed by PTCA and PDCA, and of both types of pheomelanins: benzothiazole-type and benzothiazine-type, estimated by respectively TTCA and 4-AHP not TDCA. PTCA, PDCA, TTCA and 4-AHP are the 4 major markers used to characterize the whole melanin content i.e. two eumelanin types and two pheomelanin types. TDCA is only another minor marker for benzothiazole-pheomelanin that is not taken into account.

Comment 7: Figure 17 b and its legend should be modified according to comment 5 replacing TDCA content with 4-AHP content and mentioning reference Del Bino et al Pigment Cell Melanoma Res. 2022;00:1–5

Comment 8: Line 536 "(...) different wavelengths (e.g. 350, 409, 500 nm)" 650 nm should be added in the lists of wavelength of interest.

Comment 9: Line 550 "(...) determine eumelanin, pheomelanin, and total melanin (Figure 188a, [130])": check figure number and reference number; ref 130 only contains 6 figures and only L* parameter has been measured.

Comment 10: Line 610 paragraph on other optical approaches. The authors rightly quote other approaches such as confocal microscopy, Multi-photon microscopy, near-infrared fluorescence or optical coherence tomography but omitted electro paramagnetic resonance (EPR) spectrometry that can also be used to quantify melanin content (e.g. Godeshal et al. Mol Imaging 2013 Jun;12(4):218-23).

Reviewer #3 (Remarks to the Author):

Summary:

This is an incredibly thorough review of a timely topic. To date there is no available paper that so comprehensively discusses these issues which are of great interest in the scientific community. The presentation has potential to influence thinking on this topic and future research agendas. Overall, connecting the review more clearly to current requirements for skin color diversity/measurement for pulse ox regulatory guidance/standards would be useful. Please find my additional comments below.

Section 1, Introduction:

Line 27 - disparities in performance due to skin color (not just race) go back nearly 40 years (while disparities in health/care are much more recent). Might consider including more historical context for completeness of this review. It also may be worth highlighting when the FDA started requiring skin color to be considered/accounted for, and how

Would be useful to add near Line 40-41, for which other devices does FDA and/or ISO require skin color diversity to be considered/measured during device validation

Line 40-41 - could possibly add parenthesis to expand what each of these devices types is used for

Line 52 - might note somehow that even when Fitzpatrick was being 'used' it was often not being used in its intended way (ie people often rating subjectively on a scale of 1-6 light to dark)

Line 53 - might specify that 'more recent studies' refers to more recent clinical studies of pulse oximeters in the clinical settings. And might also note where you refer to laboratory studies... it might be worth expanding for the reader about where lab studies fit into the device to market story and how skin color is defined/required to be defined and by whom FDA vs ISO etc

Section 2:

Figure 1 - Could these two images be merged in some way so the reader does not need to go back and forth? It is a nice visual

Line 121 - the number of melanocytes can be different, normal variation (e.g. Voigt-Future lines), might qualify the statement 'for the most part' 'typically'

Line 130 - might be more deliberate and cautious making generalizations/conclusions based on prior author's use/misuse of race vs ethnicity throughout

Section 3:

Line 192 - wonder if 'sun exposure' is better than 'sunburn/suntan'

Line 205 - the full history of VL scale should be included, to note its use by Nazis, lack of and lack of standardized color tiles or reference colors - making it an obsolete scale

Line 215 - worth mentioning subjective methods also subject to user bias

Section 4:

Line 230 - the term melanometers would imply a device that measure melanin; is that the best term for devices that 'measure skin color' which is determined by other factors as well, and there are devices that measure beyond melanin

Line 244 - not sure if Data acquisition techniques sub section should have a letter 'a' instead of '1'

Line 267 - might be worth a sentence on what melanin index and CIELab are (or referencing the relevant figure or section below)

Line 278 - might be worth defining higher up what you refer to as melnometers vs colorimterers etc

Line 360 - typo Figure 111

Line 373 - typo fifgure 122

Line 446 - typo "Methos"

Line 458 - ?"Error Reference source not found?"

Line 463 - might qualify that orthostatics don't change hemoglobin concentration of the blood, but blood volume of the tissue which affects hemoglobin content of the tissue volume

Line 468 - would 'color' be a better word than 'pigmentation' here

Reviewer #4 (Remarks to the Author):

Vasudevan et al. present a review of melanometry for the assessment of skin pigmentation as a means to improve pulse oximetry. The review is comprehensive and makes extensive use of the literature. This reviewer feels that some major structural changes would make the manuscript stronger, and recommends that the authors strongly consider the proposed edits:

1) Greatly shorten sections 5-8. In sections 5-6, the authors presented an analysis of correlations observed in different manuscripts with Fitzpatrick skin phototype and inter-device comparisons, but then the authors ended each section describing why the analysis is flawed overall. This was frustrating to read. The authors should just state the limitations of these types of analyses and move on to the gold standard comparison, which should really be the focus.

2) It is unclear how the authors interpret the data from papers and the various correlation coefficients. Many different measurements and correlation coefficients are provided, some being quite high, but then no critical analysis is provided on recommendations based on the papers that were reviewed. What are the authors' criteria for an acceptable melanometer? This manuscript would be strengthened considerably by the authors presenting a critique based on the reviewed data, instead of just recommending what needs to be done.

3) In general, the Figure captions should contain more text to make them more self-contained. There is a lot of information contained in each of the figures, and the captions are too terse to allow a reader to understand what is being shown in the figures.

Minor comments

1) p.3, section 2, first paragraph - the authors write "the epidermis, which contains melanin but minimal blood (Figure 1) [53, 70]. The dermis layer is highly vascularized with arterioles and venules but contains minimal melanin".

Comment: The epidermis is avascular, and the (healthy) dermis does not contain any melanin. Please edit the text.

2) p.3, section 2, first paragraph - the authors write "The thickness of epidermis and dermis layers vary strongly from site to site but tend to be in 50 – 150 μm and 1 – 4 mm ranges, respectively."

Comment: Please replace "tend" with "tends" and insert the word "the" between "in" and "50"

3) p.3, section 2, first paragraph - There is no mention of the stratum corneum in this paragraph. This is an important layer that should at least be mentioned here.

4) Figure 2 caption - What is YJBM? Is use of this figure in compliance with copyright permissions? In general, please verify that the replicated figures are used in compliance with copyright permissions.

5) p.4 - Please indicate what is meant by a "supranuclear cap"

6) p.4 - the use of the term "decay rate" is not standard when describing a spectral dependence. Please consider use of a different term.

7) p.5 - the authors write "the reduced scattering coefficient of melanin pigments also decrease"

Comment: Please replace "decrease" with "decreases"

8) p.5 - the authors mention the "spectral range of pulse oximeters"

Comment: Please provide a wavelength range in the text

9) Figure 4 - the acronym SO₂ is not described in the text

10) Figure 5 - Reflectance values are inconsistent between the two graphs. This reviewer suggests that you change one of the axes to be more similar to the other. Also, are these spectra representative ones or averages? This should be indicated in the figure caption. Finally, the acronym "FST" is used in the figure captions, but "FSP" is used in the text

11) p.6 - replace "FST" with "FSP"

12) Figure 6 - The rightmost figure should be labeled as (b), not (a). This caption would benefit from expanded text that walks the reader through the various graphics.

13) p.9 - "HbO₂" and "Hb" have not yet been defined in the text

14) p.10 - Please describe what is meant by a "standard observer" and "color matching functions"

15) p.10 - Figure 10 is referenced before Figure 9.

16) p.22 - last paragraph of Section 10 - the authors describe "high resolution optical imaging approaches".

Comment: These technologies can offer visualization of melanin at exquisite resolutions, but the one limitation is that the characterization is performed at such a small site, which might not be representative of the skin region of interest.

Reviewer #5 (Remarks to the Author):

The authors have performed a comprehensive review on the measurement of melanin in the skin by melanometry devices. The measurement by these devices is currently of high interest, because of e.g. the reported influence of skin type on pulse oximetry measurements.

The review is addressing most important issues with the measurements of melanin in the skin, and therefore has value for the scientific community. However, due to the extensiveness of the review, I have the impression that the main message of the manuscript is obscured.

1. One of the important factors in the measurement of the melanin content in the skin is the fact that it is non uniformly distributed. It is located in the thin epidermis which make the assessment of e.g. the concentration cumbersome. With visible wavelengths the influence of the underlying skin will influence the outcome of the measurement. In your reference 213, a layered model approach is used which facilitates that the influence of melanin on underlying skin measurements can be decor related. To my opinion these two or multi layered approaches are needed to improve the measurement of melanin in the skin.

2. melanin is also non uniformly distributed in the epidermis itself, which also has an influence on the measurement of melanin content in the skin. In reference your 89 this non uniform distribution has been incorporated, resulting in good fits of the measured spectra.

3. I miss in the discussion the need for layered and calibrated phantoms. These can be essential to validate device outcomes.

4. With all the knowledge that you have on this topic, I would also expect a conclusive figure with the optical properties of the epidermis. This information can be very useful for modeling optical monitoring devices by the scientific as well as commercial society.

5. There are some small typos, like reference not found, and figure numbers being 122 and 188.

Thank you for the opportunity to submit a revised draft of the manuscript “Melanometry for Objective Evaluation of Skin Pigmentation in Pulse Oximetry Studies” for publication in the Communications Medicine. We appreciate the time and effort dedicated to providing valuable feedback on our manuscript and thank the reviewers for the insightful comments. We have incorporated most of the suggestions made by the reviewers and any changes have been highlighted within the manuscript. Please find below, a point-by-point response to the reviewer comments and concerns.

Reviewers' comments:

Reviewer #1 (Remarks to the Author):

I congratulate the authors on a very thorough review of the literature for a very important and timely topic. Overall I think this review is well written and well structured. If it could be shorted a little bit I think that would help for the general reader. I do have specific comments throughout consisting of some technical points that are worth refining in my view. I also at times found that the references used to support certain statements were not accurate and would recommend the authors consider rephrasing to be more accurate.

Author response: Thank you. We have attempted to shorten the manuscript without compromising the content. We have rephrased some of our statements and verified the references used.

Introduction

- Occult hypoxemia is defined as ‘undetected low SaO₂’, but to the general reader it may be better to clearly spell out that that SpO₂ is normoxemic in the setting of SaO₂ hypoxia.

Author response: We have defined occult hypoxemia as “normoxemic SpO₂ concurrent with hypoxic SaO₂” in lines 28-29.

- Line 38 – opportunity to include recent important reference (Matthew D. Keller & Brandon Harrison-Smith & Chetan Patil & Mohammed Shahriar Arefin, 2022. "Skin colour affects the accuracy of medical oxygen sensors," Nature, Nature, vol. 610(7932), pages 449-451, October.)

Author response: We have included the reference in line 37.

- “These publications assert that such an approach will better capture skin pigmentation as a continuous variable, rather than lumping skin tones into discrete subjective categories (e.g., light, intermediate, dark), thereby facilitating enrollment of study subjects that adequately represent the full range of skin pigmentation levels.” This statement includes publication [50], which is the APSF statement. I can’t see any call for capturing skin pigmentation in a continuous fashion in that statement. If anything that statement is suggesting that there are other variables beyond pigmentation known to affect pulse ox accuracy, and that the 15% inclusion criteria should be reconsidered in light of the Sjoding paper.

Author response: The statement “These publications assert that such tools will better capture skin pigmentation as a continuous variable, rather than grouping skin tones into discrete subjective categories (e.g., light, intermediate, dark), thereby facilitating enrollment of study subjects that adequately represent the full range of skin pigmentation levels.” refers to publications – Vesoulis et. al. 2022, Okunlola et. al. 2022, Cabanas et. al. 2022 – in the previous line. The statement has been moved to include the accurate references.

Section 1 - Skin Optics

Figure 1: This diagram has issues with respect to accuracy, as it shows the stratum corneum and epidermis as having the same size? In reality the epidermis is about 5 times thicker than the stratum corneum. In Figure 2 the relative size of stratum corneum to epidermis is much more reasonable. Please replace Figure 1 skin layer schematic with something more accurate, you can check dermatology textbooks or dermatopathology textbooks.

In the diagram the depth of penetration of light has MIR called <1500 nm. That is incorrect. While some sources will extend NIR above 1000, recent work defines short wave infrared (SWIR) to 1700nm or potentially extending to 2.5 μm , and MWIR is typically 3-5 μm (with LWIR above 8 μm). It's true that around 1400nm there is a decrease in penetration depth due to water absorption peak, but penetration depth at 1300nm exceed that at 1000 nm, and that is counter to what the diagram shows. Recent work confirms this point <https://www.ncbi.nlm.nih.gov/pmc/articles/PMC5147011/>

Author response: We have replaced Figure 1 b with a skin layer schematic that is more accurate. Revised Figure 1 b only shows light penetration through skin layers for wavelengths 320 – 1000 nm because the devices that we discuss in the paper does not utilize longer wavelengths such as 1500 nm. Please find modified Figure 1 b below.

Figure 1 Overview of skin tissue optics, including schematics illustrating (a) skin anatomy [71] (Reprinted with permission from Elsevier) and (b) spectral variation in dermal light penetration [72]. (c) Extinction spectra for eumelanin and pheomelanin [73] and (d) histology images of skin samples with Fontana-Masson staining indicate variations in epidermal melanin content (epidermis in pink, melanin in black). [74] (Reprinted with permission from Photochem. Photobiol.)

“Supranuclear caps form in response to UV” -

<https://www.sciencedirect.com/science/article/pii/S0022202X15400855?via=ihub>. I think when the authors say “in highly pigmented skin, melanin often forms...” readers may think that this is only in people with highly pigmented skin. Please clarify this point.

Author response: Considering the length of the manuscript and lack of relevance of this section to the overall paper, the following content has been removed from the manuscript:

“In highly pigmented skin, melanin often forms supranuclear caps (as seen in Figure 3) superficial to keratinocyte nuclei.”

The impact of pigmentation being equivalent to SPF 13 is reported in the literature, but I’m not sure where the 70% reduction in skin cancer number comes from. Skin cancer is very rare in pigmented patients and there is no evidence to show that additional SPF protection is needed for skin cancer prevention in pigmented patients. The skin cancers that affect patient with pigmented skin occur often in areas that are not sun exposed and sadly are caught in later stages (and have higher mortality). The point about SPF and skin cancer I think is not needed for this review and should be cut.

Author response: As suggested by the reviewer, considering the length of the manuscript and due to the lack of relevance of this section to the overall paper, the following content has been removed from the manuscript:

“This level of photoprotection is similar to wearing sunscreen with SPF 13 and leads to a reduction in skin cancer rate by a factor of 70.”

I think that the typical sizes of melanosomes should be included rather than just saying that the melanosomes are larger for pigmented skin, this is especially relevant given the potential increased scattering coefficient observed in pigmented skin.

Author response: We have included typical sizes of melanosomes for white, Asian and Black skin in lines 122-124. Please see revised statement below:

“A progressive variation in melanosome size with ethnicity has also been revealed, with melanosomes in Black skin being the largest ($1.44 \pm 0.67 \mu\text{m}^2 \times 10^{-2}$) followed by Asian skin ($1.36 \pm 0.15 \mu\text{m}^2 \times 10^{-2}$) and white skin ($0.94 \pm 0.48 \mu\text{m}^2 \times 10^{-2}$) [ref].”

Figure 2:

Figure 2a quality is low – the black background with hard to read white text is not ideal. Bologna Textbook of Dermatology Edition 4 Chapter 65 Figure 65.5 has a much clearer picture illustrating the point. I have included it as an attachment for reference.

Author response: Thank you for sharing the Bologna Textbook of Dermatology Edition 4 Chapter 65 Figure 65.5 for reference. Considering the length of the manuscript, we have removed the Figure captioned “Schematic of a melanocyte in the basal layer of the epidermis transferring melanosomes to overlying keratinocytes”.

Figure 3: to those not experienced looking at the skin histology, it may have to say “basket weave” to describe the stratum corneum (i.e “..seen as thin basket weave superficial layers”)

Author response: We have removed this detail as part of an effort to shorten the manuscript.

The authors suggest that the scattering effects of melanin are too small to impact optical measurements. I encourage them to add this study to their review

<https://pubmed.ncbi.nlm.nih.gov/19687939/> where in vivo measurements were made to assess μ_a and μ_s' in subjects with varying pigmentation. It appears that there is a difference in μ_s' between light and dark subjects that is more pronounced at red than infrared wavelengths. This suggests that scattering could play a role. This would be consistent with the discussion by Chetan Patil in this work Matthew D. Keller & Brandon Harrison-Smith & Chetan Patil & Mohammed Shahriar Arefin, 2022. "Skin colour affects the accuracy of medical oxygen sensors," Nature, Nature, vol. 610(7932), pages 449-451, October.) In that op ed and in his work presented at SPIE Chetan showed that it is the inclusion of scattering properties of melanin with Monte Carlo models that is necessary to recapitulate the Sjoding results in silico. To my reading the authors come off as dismissive of scattering and focus completely on absorption. This implies that the bias is simply a signal to noise problem and not a problem of differential pathlength due to different scattering between red and IR channels for light and darkly pigmented skin.

Author response: Thank you for this suggestion. Although it would have been interesting to explore this aspect, we have not included it in the manuscript due to the relatively small magnitude of change in scattering with respect to skin type/ skin color. Please note that the reference you have provided [<https://pubmed.ncbi.nlm.nih.gov/19687939/> Tseng, S. H., et. al. (2009). Optics Express] quantifies the change between FSP V-VI versus I-II and III-IV as 1-10% (see below).

"The average reduced scattering coefficients of the three groups are very similar. The average reduced scattering coefficients of the skin type V-VI group are 1-10% higher than those of the skin type I-II and III-IV groups."

We are not aware of any peer-reviewed publication containing evidence that pigmentation-correlated changes in epidermal scattering impact pulse oximetry. We have read closely the proceedings article by Arefin et al., (Proc. SPIE 2023) yet it does not attempt to establish that inclusion of changes in scattering coefficient as a function of melanin content is necessary to simulate disparities in measurement accuracy.

Pigmentation-dependent variations in performance have been shown in several optical technologies, including photoacoustic imaging, bilirubinometry, hyperspectral imaging and cerebral oximetry. All of the plausible explanations for these impacts (as well as the correction algorithm used for bilirubinometry), are based on melanin absorption, and it does not appear reasonable from a tissue optics perspective that these effects are due to changes in epidermal scattering. Furthermore, the strong, spectrally dependent reduction in detected reflectance signals with skin color [e.g., Mendenhall et al. Appl Opt 2015] clearly aligns with the known absorption spectrum of melanin.

As regards pulse oximetry, the available evidence indicates that an absorption-based mechanism is more plausible than one based on scattering. There are several potential mechanisms: high absorption near the tissue surface may modify the optical pathway of detected light; a red-shift in the 660 nm LED centroid may be caused by melanin [Bickler Anesthes 2023], or low signal intensity combined with low perfusion index (see Gudelunas et al., Medarxiv 2023) may make it more difficult for the oximeter to differentiate between the absorption effects of melanin and deoxy-Hb.

Please note that we have included the following text (highlighted) in lines 145-147 to emphasize that scattering might play a role in the pulse oximeter performance bias.

“Although scattering may play a role in pulse oximetry racial disparities, given the minimal thickness of the epidermis and relatively small magnitude of change in scattering, the impact of this effect is likely not significant.”

For the description of FPS scale, I think it is useful to know that it was designed for the purposes of predicting what UV dose could be given safely to white patients undergoing systemic treatment with psoralen for their psoriasis. It was necessary because estimating a safe dose was deemed not possible to be done by just looking at someone’s apparent pigmentation, and instead required a survey regarding the propensity to tan and/or burn.

Author response: As suggested by the reviewer, we have included the application for which Fitzpatrick Skin Phototype was designed in the following statement (lines 170-173).

“FSP (I – IV) was originally developed in 1975 [ref] as the “Fitzpatrick-Pathak skin typing system” to assess ultraviolet light sensitivity in “persons with white skin” to select correct initial UVA dose for patients undergoing oral methoxsalen photochemotherapy for psoriasis.”

Figure 6a is an excellent figure to include, but I think the purpose of that figure in the perspective that Fitzpatrick wrote is to show that a Fitzpatrick skin type is defined by the change in skin with sun exposure – Type 1-4 are indistinguishable from facultative pigmentation alone and can only be differentiated when considering the skin tone after sun exposure. The statement “An FST skin type can correspond to different skin colors depending on whether constitutive or facultative pigmentation is present” implies that there is utility to defining an FST visual scale based on constitutive pigmentation, but the Figure clearly demonstrates that there is no difference (and therefore no information to be gleaned) for facultative FST 1-4.

Author response: The statement has been removed.

I would strengthen this statement “While skin color charts based on the FSP method have been used [104, 105], these represent a non-standard implementation that contradicts the original intent of FSP to assess differences in sensitivity to sunburn for individuals with similar skin color”. One challenge in particular is that there is no established color palette for the FSP I-VI categories. These categories have not been mapped to a color space, and their standardization for what colors are used by various studies that report enrolling subjects based on Fitzpatrick type.

Author response: We agree with the reviewer’s suggestion and have included more content to strengthen the statement (lines 180-182).

“While perceived FSP has been determined via the use of skin color charts [ref], there is no established color palette for perceived FSP I-VI categories, and they have not been mapped to a standardized color space.”

In general FSP is used ubiquitously as a visual scale despite the fact that it was specifically designed to measure something that was not assessable for dermatologists by eye. To make clear this problem, I suggest that the authors call any study that uses FSP from visual estimation a ‘perceived FSP’ rather than an FSP that is defined as originally defined through a set of questions about skin reactivity.

Author response: We have used “perceived FSP” to defined FSP determined visually. We have also revised Figure 5 a that shows compiled correlation results between melanometers and FSP to include different color markers for different FSP types.

I know this may be a controversial point and one that the authors want to avoid bringing up, but the Von Luschan tiles are on display in the Holocaust museum as they were used by the Nazi's to characterize skin color in racial studies <https://collections.ushmm.org/search/catalog/irn564926> and Von Luschan was also a member of the German Society for Racial Hygiene. At the same time he appears to have opposed scientific racism in his work <https://www.jstor.org/stable/41157700>. I don't know if it's possible to bring all this up and it's not relevant to the review, however not mentioning the complicated past of Von Luschan's tiles may serve to alienate some readers. Is there something the authors can say to put this in appropriate context?

Author response: We are aware of these issues regarding von Luschan, yet we believe this is beyond the scientific scope of our manuscript. Based on our reading of the literature, it appears that von Luschan intended the scale as a tool to help "annihilate the thesis of race inferiority" (as DuBois described von Luschan's work). The co-opting of a skin color scale for nefarious purposes could be done with almost any scale (which perhaps should serve as a caution for modern scientists). That said, we also acknowledge various unsettling aspects of von Luschan's research unrelated to the color scale and have added text in Lines 188-190 to help direct readers to academic sources that provide additional context on this controversial topic.

"VLS was extensively used in anthropological field research in the 1950s [ref], but the legacy of von Luschan and VLS has been considered controversial [Smith, John David. *Amerikastudien/American Studies* (2002), Beatty, Joel Scott. Diss. Michigan Technological University (2017), Foreman, Deirdre. Diss. Rutgers University-Graduate School-Newark (2017)]."

Line 221-224 are confusing: "Subjective methods such as FSP use a relatively coarse categorization for a continuum of skin tones (fair to dark) [120] which causes individual categories such as FSP VI to cover a wide range of melanin concentrations (and reflectance values, as shown in Figure 5)". I think the authors are making the point that a visual perceived FST fails to capture details regarding how skin pigmentation changes in response to sun exposure. However if FST is measured appropriately with questions about skin reactivity and not by color matching then information about how skin changes in response to sun exposure is captured. The authors reference this paper <https://jamanetwork.com/journals/jamadermatology/fullarticle/1737180> and in Table 3 there is a nice summary of proposed questions to improve FST, but these questions are not visual color matching based and are in my view in the spirit of what FST was designed to do.

Author response: Our intent was not to address response to sun exposure, rather to highlight that 'perceived FSP' does not provide an accurate measure of pigmentation for subjects with dark skin.

I appreciate the authors include the work of Pichon et al (ref 125), but I think a key message of that paper was that the questions about sun burning and tanning that were designed for white people were difficult to interpret and not relevant to people with type V or VI skin. So in that paper it wasn't physicians being unreliable in assigning FSP, but it was that the questions themselves made self-assignment by subjects unreliable. This is an important point about the inequities that can arise in medical tools when the tools are not built to serve a diverse population, and I think is in line with the points that underpin the need for this review in the first place.

Author response: We have modified our description of the Pichon study accordingly. Please find revised statement in Lines 202-203:

"Lack of reliability can also be caused when FSP – an approach developed for light-skin subjects – is implemented to study a diverse population (Pichon et. al.)."

Section 4:

Table 1. In this table the XRite spectrophotometer is not listed, despite it being the spectrophotometer of choice in Figure 5. I know not every device can be listed but given it's presence in Figure 5 it seems appropriate to include. Also the CM700d is listed as having spectral range from 300-700. I think this is an error, the range is 400-700 ()

Author response: We have corrected the wavelength range of CM700d in Table 1. We have also included XRite MA68II spectrophotometer in Table 1.

CM700d (Konica Minolta)	Reflectance spectroscopy	400-700	Melanin conc. ^a MI ¹ ^a L*a*b*
MA68II (X-Rite)	Reflectance spectroscopy	400-700	L*a*b*

In the text the authors state that examples of tristimulus colorimeters can be found in Table 1, but in Table 1 they also describe devices as having 'tristimulus colorimetry' applied during processing. I think using 'tristimulus colorimetry' to describe the processing step that occurs within the device is potentially confusing to non-expert readers. In Reference 66 from Ly et al they take a different approach – they call devices like the Delfin a tristimulus colorimeter. In numerous publications and device descriptions I've seen I also see the term colorimeter to mean a device that is designed to measure the color that is perceived by the human eye. Ly et al basically call a tristimulus colorimeter a type of device that measures spectral response across 3 narrow bands corresponding to the bands that the human eye perceives.

It is true that a spectrophotometer uses principles of tristimulus colorimetry to extract tristimulus values from a more complete reflectance spectrum. However I like the approach of Ly et al to make clear that the wide band spectrophotometer contains more rich information than a narrow band spectrophotometer or tristimulus colorimeter, and that tristimulus values can be mathematically extracted from the full spectrum. In fact for a broadband spectrophotometer one can apply the color matching equations for an arbitrary illuminant and observer angle, deriving different tristimulus values for each given illuminant and observer. You can not do that with a tristimulus colorimeter which typically only has one illuminant. For this reason I think the term tristimulus colorimeter should be reserved in Table 1 to the RGB spectral reflectance devices like the Delfin.

Author response: We have reformatted Table 1 to include device type instead of acquisition and processing columns. We have also reserved the use of 'tristimulus colorimeter' to define RGB spectral reflectance devices such as Delfin, CL400 etc.

Device	Device Type	Wavelength (nm)	Metrics Used
Mexameter MX16 (Courage-Khazaka)	Narrowband reflectance	568,660,880	MI ¹ = 500/log ₁₀ (5) × [log ₁₀ (R _{880nm} /R _{660nm}) + log ₁₀ (5)] EI ² = 500/log ₁₀ (5) × [log ₁₀ (R _{660nm} /R _{568nm}) + log ₁₀ (5)] [ref]
Mexameter MX18 (Courage-Khazaka)	Narrowband reflectance	568,660,870	MI ¹ = log ₁₀ (R _{870nm} /R _{660nm}) × 1000 EI ² = log ₁₀ (R _{660nm} /R _{568nm}) × 1000 [ref]

CL400 (Courage-Khazaka)	Tristimulus colorimeter	440-670	L*a*b*,ITA
DSM, DSM II/III (Cortex Technology)	Narrowband reflectance	568,655	$MI^1 = \log_{10}(1/R_{655nm}) \times 100$ $EI^2 = \log_{10}(R_{655nm}/R_{568nm}) \times 100$ [ref]
DSMII/III (Cortex Technology)	Tristimulus colorimeter	620,530,460	L*a*b*
CM700d (Konica Minolta)	Reflectance spectroscopy	400-700	Melanin conc. ^a MI^1 ^a L*a*b*
CM-2002 (Konica Minolta)	Reflectance spectroscopy	400-700	Melanin conc. ^a L*a*b*
CR-200/300 (Konica Minolta)	Tristimulus colorimeter	450,550,600	L*a*b*
CR-221-R (Konica Minolta)	Tristimulus colorimeter	450,560,600	L*a*b*
SkinColorCatch (Delfin Technologies)	Narrowband reflectance Tristimulus colorimeter	620,540,460	MI^1 , EI^2 L*a*b*,ITA
USB2000+ (Ocean Insight)	Reflectance spectroscopy	300-1100	Melanin conc. ^a
S2000 (Ocean Insight)	Reflectance spectroscopy	350-1100	AUIC ³ 450-615nm ^a
CHECK3 (Datacolor)	Reflectance spectroscopy	400-700	L*a*b*, ITA ^a
MA68II (X-Rite)	Reflectance spectroscopy	400-700	L*a*b*

One thing that is missing from Table 1 is a mention of whether devices include or exclude specular reflection (some allow one to choose, others give no choice), and also what the aperture and illuminant is for various devices. The CM700d has different aperture sizes, given that there is interest in determining a standard approach for melanometry, I suggest the authors include information comparing differences between device apertures or source-detector distances. These are variables that could affect the device output and therefore could have an impact on estimates of melanin content.

Author response: Given the need to reduce the manuscript length, it does not seem feasible to add more device details.

Figure 8 – It is good to show the different instruments, but a scale bar or at least showing them all in the same view would be helpful. The device in Figure 8e is much more compact than the device in 8d but you wouldn't know by looking at the figure.

Author response: We have retaken the picture with all the devices in the same view (Figure 4 b in the revised manuscript version).

Figure 10 – I could not find the color matching functions in 10a within reference 66 or 162. I think a better graph for that comes from Konica Minolta’s website

<https://www.konicaminolta.com/instruments/knowledge/light/concepts/07.html#:~:text=4.1>

TRISTIMULUS COLORIMETRY, of these three primary colours. Or this is a resource that the authors may find interesting to share or review – it has some example code for implementing the conversion of a spectrum to an XYZ tristimulus value set and also shows the color matching functions.

<https://www.fourmilab.ch/documents/specrend/>

Author response: The color matching functions figure was generated by us and hence not found in the references.

Figure 10c is reprinted from multiple papers, but I think the equation in the bottom is written in a confusing fashion : $ITA^\circ = (ATAN(L^*-50)/b^*)...$... This way of writing the equation does not make clear that you are taking the ATAN of the ratio (i.e. $ATAN[(L^*-50)/b^*]$). While the authors do provide the correct equation in the manuscript, I think including the figure with this confusingly written equation could perpetuate errors.

Author response: We have removed the equation from Figure 4f (former Figure 10c).

The authors make a good point about ITA bins. “However, it should be noted that the ITA skin color bins are non-uniform in width with much wider bin size for darker skin compared to lighter skin.” However, I think the caution should be expanded, because just as was the case with FST, type I-IV ITA span the range for Caucasian people, and V and VI are left for all people with pigmented skin. These bins have been used in multiple studies but I think it is important to clarify their history just as they clarified the history of the FST bins. The initial definition by Chardon in 1991 had 4 bins, 10° as the smallest bin cutoff and only included Caucasian subjects with skin varying from very light to tan, and $<10^\circ$ was considered brown. Only about 20 years later did the Del Bino work introduce the -30° cutoff for ‘dark’. Just as darkly pigmented skin was an afterthought for FST, so too was it for the ITA bins.

Author response: An expansion of this discussion has been made (lines 303-309) to address ITA and ITA-binned skin color categories from the Chardon article.

“Early work on ITA [Chardon 1991] was limited to values above 10° , and these values were used to define four primary categories: very light, light, intermediate, and tan. Subsequent studies considered ITA values as low as -90° and added “brown” and “dark” categories (Error! Reference source not found. f) [Del Bino]. However, the ITA skin color “bins” for each category are non-uniform in width, with much

wider bin size for darker skin compared to lighter skin. To develop ITA-based categories more closely corresponding to differences in epidermal melanin content, a strategy employing uniform bin sizes may be more appropriate.”

Section 5:

One potential weakness of prior studies that compared FST to melanometer-derived values is that the application of FST is inconsistent – sometimes it is a perceived FST based on color matching to a non-standardized FST color palette, and sometimes it is based on asking questions about skin response to UV. In Figure 11 it would be good to know whether the studies listed used perceived FSP or questionnaire defined FSP – do the cases where correlation is strongest represent cases where it was perceived FST? Indeed the “observer rated pigmentation scales” had an impressive correlation coefficient of $R=0.92$, so perhaps FST agrees better with melanometry when FST is used as a visual scale.

Author response: We have modified Figure 5 a (former Figure 11) to include markers for the method used to determine the FSP types. We have also included the following text in Section 5 paragraph 3 (lines 340-345).

“Different methods have been employed to determine FSP (marker colors in Figure 2 (a) Correlation coefficients between melanometer outputs and FSP scale for several commercial devices [ref]. Marker colors represent different methods employed to determine FSP [black – questionnaire, blue – perceived FSP, red – perceived FSP plus questionnaire]; Correlation between FSP scale and (b) Mexameter MX18 MI [ref] (c) CM2600d ITA [ref]; (d) MI based FSP classification [ref]

a). FSP was determined using questionnaires in 8/11 studies (20/26 correlation results), using visual perception in 1/11 studies (1/26 correlation results), and using both visual perception and questionnaires in 2/11 studies (5/26 correlation results) . While the lone study using visual perception of FSP alone reported strong correlation ($R = 0.886$, $p<0.001$) between FSP and melanometer output, [Isa Zaleha Md MJPHM], further comparative analysis between different FSP methods could not be performed due to limited studies.”

Figure 2 (a) Correlation coefficients between melanometer outputs and FSP scale for several commercial devices [ref]. Marker colors represent different methods employed to determine FSP [black – questionnaire, blue – perceived FSP, red – perceived FSP plus questionnaire]; Correlation between FSP scale and (b) Mexameter MX18 MI [ref] (c) CM2600d ITA [ref]; (d) MI based FSP classification [ref]

I think the recent paper by Osto et al is quite excellent as a reference. In this paper FST was determined by dermatologists in a photomedicine unit, meaning that it was used as originally intended, and ITA was measured by spectrophotometer. The figure from this paper is better than the figure in 11c which doesn't include Type 1 and comes from a paper whose primary goal was not to determine how FST and ITA relate. Also the paper from 11c does not describe how FST was determined.

Author response: We have modified Figure 5 c to include FSP vs CM2600d ITA data from Osto et. al. We also calculated the correlation coefficient for this data ($r = -0.9110$) and have included it in the compiled dataset in Figure 5 a. Please see figure included in the response above.

Minor points: Line 360, 364, 366 Figure 11 is listed as Figure 111. Line 373 lists Figure 12 as Figure 122.

Author response: The error in Figure numbers has been corrected.

Figure 12: There are multiple errors in Figure 12b. First of all, the categories are reversed. The 'very light' category is put in the $< -30^\circ$ bin but should be in the $> 55^\circ$ bin. The order of the y-labels must have gotten switched accidentally. Also the Chardon 1991 paper had no -30° bin, that was only 20 years later by Del Bino. I'm not sure what the purpose is of Figure 12b though, it just shows a visual representation of the ITA bins which were already described in the manuscript early. Is the goal to show that the ITA bins mirror the MI bins in the Wilkes study? That may be the authors listed the ITA bins light to dark to match

the FST 1-6, but this creates a problem because ITA decreases as MI increases, so you won't be able to easily make the graphs look the same.

Author response: We agree that the visual representation of the ITA bins was already demonstrated in Figure 4 f and hence have removed this figure.

Section 7:

Good job summarizing lots of data and studies. One thing that is not clear from this data though is whether the results apply to subjects with dark pigmentation. Given that these subjects are often not well represented in studies, I think it is useful to know if these subjects are represented in these melanometer validation studies. The DSMII study for example (Ref 182) has 82% Caucasian subjects with 0 subjects with type VI FST. I am not sure about the other studies included in Figure 14, but I think it's important for the authors to look into that to provide some sense of how applicable these validation studies are for skin of color patients.

Author response: In the compiled repeatability dataset presented, 5 out of 13 studies reported the FSP distribution of the study population, amongst which only 2 studies included subjects of FSP VI skin type. We have included more details to this section (lines 394-399):

“The heterogeneity of skin pigmentation of the sampled population varies between different studies. In the compiled repeatability dataset presented here (**Error! Reference source not found.**), 5 out of 13 studies reported the FSP distribution of the study population, of which only 2 studies included subjects of FSP VI. For the rest of studies, 5 studies reported race and 3 studies did not report any information on patient skin type or ethno-racial background. No significant trend was observed in the repeatability data for studies that included FSP VI versus studies that did not include FSP VI.”

Section 8:

Line 457 – missing reference

Author response: Reference to Figure 8 has been included.

Section 9:

Few typos (Line 509 Figure is called Figure 122b, Lin 529 Figure is called Figure 177, and many other examples where the 1s digit is repeated in calling a figure), but otherwise this was an excellent section and one of the strongest parts.

Author response: The error in Figure numbers has been corrected.

Section 10:

The authors do an excellent job finding relevant literature, but could improve the text by being more accurate. Reference 201 is described as “reflectance spectrum mean slope over 620 to 720 nm”. That is incorrect however, because they say: “By fitting the logarithm of the ratio of the remittance spectrum from vitiligo involved skin to normal skin of the same volunteer, from adjacent sites, we obtained a straight line fit to the experimental data”. So in reality they took the log of the reflectance spectrum and found the slope of that over 620 to 720. In essence this is a ratio of reflectance at 720 to reflectance at 620.

And in reference 202 for example the authors described it as a 'difference between reflectance at 650 and 710'. This is not accurate either, in that work they define melanin index as follows "Specifically, the melanin index is defined by the slope of OD (λ) in the region above 620-640 nm (Fig.2): $M=100(OD_{650} - OD_{700})$ ". It is a difference of the optical density, which is the log of reflectance, so it is in reality a ratio of reflectance at 700 and 650 since a difference of a log is equal to the log of the ratio. The work by Dawson (ref 138) is also a difference of the log of reflectance at 650 and 700, so it is identical to the approach described in 202 and is also mentioned in ref 201.

Indeed all of these approaches are just a slope of the linear absorbance spectrum (the log of reflectance) above 650, which is one of the methods the authors already described. So while the authors describe this work as difference from section 4, it is identical to what was previously described. This is an easy mistake to make (missing that a logarithm is applied), but critical that the authors go back and ensure that alternative methods described in section 10 are indeed different.

Author response: We agree that a couple of the sentences in Section 10 are essentially redundant with Section 4. Considering the length of the manuscript we have removed a few of these sentences from Section 10.

Section 11:

One of the issues that the authors do not address is the practical aspect of measuring the nail bed. Many devices are so bulky that it is hard to position reliably at the nail bed, or the devices have measurement windows that are larger than the size of a nailbed. This can lead to an otherwise reliable device being very unreliable due to challenges with the site of measurement.

Author response: To address this point we have added text (lines 622-632) "Another practical issue for measuring finger sites includes the challenge of measuring small regions with bulky instruments and devices that have a large field of view."

"However, since pulse oximeters are commonly used on the index finger, the range of pigmentation levels can be expected be relatively low [210]" The authors here mean the palmar side of the finger right? The 210 reference used SFDI and measured at the palm and forearm – there were no measurements done at the dorsal finger. The dorsal finger is a sun exposed site so should have similar pigmentation to forehead. Is it the case that the pulse ox signal only goes through the nail bed and doesn't go through the finger beyond the eponychium? It is hard to know exactly where a pulse oximeter is positioned on a patient, but this is an x-ray that may be instructive. The sensor position may be between the DPI and the tip of the distal phalangeal bone, wouldn't that be close to the eponychium and therefore include skin beyond the nail bed?

Author response: For clarification, we have added text to mention "palmar surface and nail" in lines 608-609:

"However, pulse oximeters are commonly used on regions of the index finger (palmar surface and nail) where the range of pigmentation levels is relatively small"

In the conclusion too the authors suggested that the range of pigmentation is small at the measurement site and I'm not sure we know this because the dorsal finger likely is part of the pulse ox measurement site.

This text does not exclude the possibility that other sites (dorsal skin) might be measured on the finger, rather it points out that there is a good chance that researchers will want devices that are accurate enough to measure low-pigmentation sites on the finger: "...higher performance spectroscopy systems appear to enable the greatest accuracy and flexibility. The latter may also be particularly useful for pulse oximetry studies that address the low pigmentation levels in the distal finger."

Author response: Please see response to the previous comment.

Reviewer #2 (Remarks to the Author):

To clarify the potential for disparities in pulse oximeter accuracy used in healthcare and due to evidence of racial disparities especially in patients with dark skin, the manuscript addresses the need for objective methods to quantify melanin content in skin. After an introduction on the optics of skin pigmentation, the authors review the literature on melanometry and commercially available devices used as tools for objective skin pigmentation assessment. They compare the melanometry approach to subjective skin color classification methods i.e. self-reported skin color, race/ethnicity or other subjective methods. They also compare melanometry to gold standard melanin quantification methods i.e. Fontana-Masson staining, HPLC, spectrophotometry. The authors also discuss further directions that could improve these reflectance-based techniques and finally give the best practices and standardization that could enhance the rigor and quality of clinical studies on pulse oximetry and other optical technologies.

Overall impression of the manuscript:

The manuscript is interesting, well written, pleasant to read. It provides quantitative analysis data from the literature to assess the effectiveness of melanometer approach, giving the advantages and drawbacks of different commercially available devices and their usage validity in comparison to other objective quantification methods as well as subjective ones. The article identifies the best practices for standardized melanometer implementation and gives future directions to optimize the accuracy of melanin quantification.

This work really brings something to clinicians and researchers, not only for pulse oximetry studies. Even if racial bias in the latter is the starting point for this work, I wonder if "Melanometry for Objective Evaluation of Skin Pigmentation" wouldn't be a more appropriate title for the manuscript.

Author response: Thank you for the suggestion. As the motivation for this work is racial disparities in pulse oximeters and the manuscript introduction, discussion and conclusion leans towards the feasibility of using melanometers in pulse oximeter clinical studies we intend to keep the original title – "Melanometry for Objective Evaluation of Skin Pigmentation in Pulse Oximetry Studies"

Comments for transmission to authors:

Comment 1: Line 123 "(...) whereas Asian skin contains approx. 2-fold higher melanin than Caucasian skin". Authors should qualify these statements and stress that classifications based only on race/ethnic origin and not on skin pigmentation could be misleading. The reader should not take these data for granted. Skin from donors of Caucasian and Asian descent with the same phototype or ITA° have the same (eu)melanin content. Furthermore, Asian skin comprises skin of individuals from Northeast Asia (Japan, Korea, China...) and Southeast Asia (India, south of China, Indonesia, Malesia...). So the 2-fold

higher melanin content might be true when comparisons are done between European skin and Indian skin for example but not when comparing European to Japanese, North Chinese skin etc. Furthermore, in each country there is a variability in skin tones as described in Figure 2 of Del Bino and Bernerd British Journal of Dermatology (2013) 169 (Suppl. 3), pp33–40. So when it comes to Asian skin it should be specified from which geographic area and of which color.

Author response: We agree and have added text as a caveat just following these statements, as well as a citation to Del Bino's 2013 paper (lines 117-118).

"However, extensive variations in melanin content exist within these groups, including by country [Del Bino et. al. BJD 2013]."

Comment 2: Table 1 The last line on CHECK3 comes after the legend: this should be inverted.

Author response: This has been rectified.

Comment 3: Figure 6 is blurred, can you increase the quality?

Author response: We have attempted to sharpen this image. Unfortunately, we could not acquire a high-resolution version of this image.

Comment 4: Line 397 "However, a few studies suggested the MI appears to be a better indicator of melanin compared to L*, particularly due to decreased correlation observed between MI and L* in more vascularized body sites with high intra-individual variability" : "more vascularized" or "darker skin tones"? L* represents the level of grey the lightness of the skin while b* represent the level of yellow or blue of the skin. For objective skin color assessment with all the devices described it should be recommended to avoid measuring uneven skin areas i.e. blood vessels, nevi, stretch marks etc. Furthermore "MI appears to be a better indicator of melanin compared to L*" : this is logical because melanin content is different from skin color and L*b* and ITA ° measures skin color and not melanin content.

Author response: Indeed, the Shriver paper does mention that in more vascularized tissue (not darker skin) L* will likely show less robustness to blood content than MI does. ("In lightly pigmented persons, differences in degree of vascularization of different body sites make the use of L* troublesome"). This is likely because L* is less inherently robust to blood since it depends on reflectance at shorter wavelengths where hemoglobin absorption coefficient is very high. MI, however, is typically based on longer wavelengths where blood is much less absorbing. Explanatory text has been added to this paragraph.

The following text has been added in Section 11 (lines 596-598): " Additionally, measurements should avoid highly heterogeneous regions – such as those containing visible blood vessels, nevi, and scars – as well as regions with discolorations that may be endogenous or exogenous (e.g., tattoos), or those to which cosmetics or medications were applied."

Comment 5: Line 529 "Figure 177 and b" should be Figure 17

Author response: The error in Figure numbers has been corrected.

Comment 6: Line 529 "Figure 17 a and b illustrate a typical HPLC chromatogram for dark human skin and the contribution of eumelanin (PTCA, PDCA) and pheomelanin (TTCA, TDCA) (...)". The latter should be

(TTCA, 4-AHP) not (TTCA, TDCA) as illustrated in Figure 3a of Del Bino et al Pigment Cell Melanoma Res. 2022;00:1–5 that shows the contribution of both types of eumelanins : DHI-CA eumelanin and DHI-eumelanin, assessed by PTCA and PDCA, and of both types of pheomelanins: benzothiazole-type and benzothiazine-type, estimated by respectively TTCA and 4-AHP not TDCA. PTCA, PDCA, TTCA and 4-AHP are the 4 major markers used to characterize the whole melanin content i.e. two eumelanin types and two pheomelanin types. TDCA is only another minor marker for benzothiazole-pheomelanin that is not taken into account.

Author response: We have modified Figure 9 c (former Figure 17 b) to demonstrate contributions of eumelanin (PTCA and PDCA) and pheomelanin (TTCA and 4-AHP) in the different skin color groups and have modified the text accordingly:

“**Error! Reference source not found.** illustrates a typical HPLC chromatogram for dark human epidermis and **Error! Reference source not found.** demonstrates the contribution of eumelanin (PTCA, PDCA) and pheomelanin (TTCA, 4-AHP) in 6 skin color groups (very light to dark) based on ITA classification.”

(c) the levels of eumelanin (PTCA, PDCA) and pheomelanin (TTCA, 4-AHP) markers in different skin color groups [ref]

Comment 7: Figure 17 b and its legend should be modified according to comment 5 replacing TDCA content with 4-AHP content and mentioning reference Del Bino et al Pigment Cell Melanoma Res. 2022;00:1–5

Author response: We have modified Figure 9 c (former Figure 17 b) to demonstrate contributions of eumelanin (PTCA and PDCA) and pheomelanin (TTCA and 4-AHP) in the different skin color groups and have modified the text accordingly. Please see response to comment 6 above.

Comment 8: Line 536 “(...) different wavelengths (e.g. 350, 409, 500 nm)” 650 nm should be added in the lists of wavelength of interest.

Author response: The text has been added in lines 501-503.

“Melanin content can also be estimated spectrophotometrically using optical absorption methods at different wavelengths (e.g., 350, 409, 500, 650 nm) after solubilization of melanin from tissue samples in NaOH or Soluene-350 [ref].”

Comment 9: Line 550 “(...) determine eumelanin, pheomelanin, and total melanin (Figure 188a, [130])”: check figure number and reference number; ref 130 only contains 6 figures and only L* parameter has been measured.

Author response: The mention of the figure was an error and has been removed. The correlation coefficients of b* with eumelanin, pheomelanin and total melanin were derived from Table 5 in reference [Matsunaka, H., et. al. (2017). Skin research and technology]. We have clarified the reference in the text as follows (lines 514-516).

“It can be observed that b* did not demonstrate a significant correlation with HPLC determined eumelanin, pheomelanin and total melanin (Table 5 in Error! Reference source not found.Matsunaka, H., et. al. (2017). Skin research and technology).”

Comment 10: Line 610 paragraph on other optical approaches. The authors rightly quote other approaches such as confocal microscopy, Multi-photon microscopy, near-infrared fluorescence or optical coherence tomography but omitted electro paramagnetic resonance (EPR) spectrometry that can also be used to quantify melanin content (e.g. Godeshal et al. Mol Imaging 2013 Jun;12(4):218-23).

Author response: We have included the following text and referenced the Godechal, Quentin, et al. paper.

“Electron paramagnetic resonance (EPR) spectrometry can detect melanin pigments present in skin, hair, and most types of the malignant melanomas [Godechal, Quentin, et al. Molecular imaging 12.4 (2013): 7290-2012]”

Reviewer #3 (Remarks to the Author):

Summary:

This is an incredibly thorough review of a timely topic. To date there is no available paper that so comprehensively discusses these issues which are of great interest in the scientific community. The presentation has potential to influence thinking on this topic and future research agendas. Overall, connecting the review more clearly to current requirements for skin color diversity/measurement for pulse ox regulatory guidance/standards would be useful. Please find my additional comments below.

Author response: Thank you. We revised section 11 – best practices and standardization to include more details on consensus guidelines for skin pigmentation measurements.

Section 1, Introduction:

Line 27 - disparities in performance due to skin color (not just race) go back nearly 40 years (while disparities in health/care are much more recent). Might consider including more historical context for completeness of this review. It also may be worth highlighting when the FDA started requiring skin color to be considered/accounted for, and how

Author response: While it would be interesting to add additional historical context, the manuscript is currently overlength and the references provided should be sufficient to establish the justification for our work.

Would be useful to add near Line 40-41, for which other devices does FDA and/or ISO require skin color diversity to be considered/measured during device validation

Author response: We are not aware of any other devices for which such information is specifically required, although FDA has published guidance documents addressing the need for diversity and inclusion in evaluating device safety and performance.

Line 40-41 - could possibly add parenthesis to expand what each of these devices types is used for

Author response: While it would be interesting to expand on application areas of other optical technologies impacted by skin pigmentation, this is beyond the scope of this paper and the manuscript is currently overlength.

Line 52 - might note somehow that even when Fitzpatrick was being 'used' it was often not being used in its intended way (ie people often rating subjectively on a scale of 1-6 light to dark)

Author response: In order to clarify how FSP has been assessed in different ways, we have used "perceived FSP" to defined FSP determined visually. We have also revised Figure 5 a that shows compiled correlation results between melanometers and FSP to include different color markers for different FSP types.

Line 53 - might specify that 'more recent studies' refers to more recent clinical studies of pulse oximeters in the clinical settings. And might also note where you refer to laboratory studies... it might be worth expanding for the reader about where lab studies fit into the device to market story and how skin color is defined/required to be defined and by whom FDA vs ISO etc

Author response: While it would be interesting to expand on these areas, this is beyond the scope of this paper and the manuscript is currently overlength.

Section 2:

Figure 1 - Could these two images be merged in some way so the reader does not need to go back and forth? It is a nice visual

Author response: We agree with the reviewer and have merged the first few figures to form Figure 1.

Figure 3 Overview of skin tissue optics, including schematics illustrating (a) skin anatomy [71] (Reprinted with permission from Elsevier) and (b) spectral variation in dermal light penetration [72]. (c) Extinction spectra for eumelanin and pheomelanin [73] and (d) histology images of skin samples with Fontana-Masson staining indicate variations in epidermal melanin content (epidermis in pink, melanin in black). [74] (Reprinted with permission from Photochem. Photobiol.)

Line 121 - the number of melanocytes can be different, normal variation (e.g. Voigt-Future lines), might qualify the statement 'for the most part' 'typically'

Author response: We have modified the statement as follows (lines 113-115)

"Skin color is affected by size, number, and shape of melanosomes, although for the most part the number of melanocytes in a given anatomical site is more consistent with varying pigmentation"

Line 130 - might be more deliberate and cautious making generalizations/conclusions based on prior author's use/misuse of race vs ethnicity throughout

Author response: We have included the caveat that race/ethnicity does not correlate with melanin content. Please see below lines 120-122

"Histologically detected melanin content show considerable differences for different ethnicities (white, Asian and Black - Figure 1 d) [ref] with the caveat that ethn racial categories do not correlate with the epidermal melanin content."

Section 3:

Line 192 - wonder if 'sun exposure' is better than 'sunburn/suntan'

Author response: We have reworded the statement as suggested in lines 176-178.

“It is worth noting that skin color can refer to constitutive pigmentation – an individual’s inherent pigmentation in the absence of solar/ultraviolet exposure, which was originally relevant to FSP – or facultative pigmentation – which accounts for changes due to sun exposure.”

Line 205 - the full history of VL scale should be included, to note its use by Nazis, lack of and lack of standardized color tiles or reference colors - making it an obsolete scale

Author response: These issues were addressed under comments by Reviewer #1 and includes references to papers that discuss von Luschan’s contributions and controversy. We would encourage the reviewer to read some of these references for a more thorough understanding of von Luschan’s work.

Line 215 - worth mentioning subjective methods also subject to user bias

Author response: We have included user bias in the statement in lines 197-199.

“Although, subjective methods have been used for categorization of skin pigmentation levels for many years and the aforementioned methods provide a general basis for skin pigment classification, they are limited due to errors resulting from observer/user bias, lighting conditions, and skewed self-reporting [ref].”

Section 4:

Line 230 - the term melanometers would imply a device that measure melanin; is that the best term for devices that ‘measure skin color’ which is determined by other factors as well, and there are devices that measure beyond melanin

Author response: While colorimeters may indeed measure skin color, researchers studying racial disparities in optical device performance have an interest in these devices due to their ability to calculate ITA, which is strongly correlated with melanin content. Furthermore, devices that measure melanin index implement algorithms developed for estimating melanin content. While perhaps not ideal, we believe this term is useful.

Line 244 - not sure if Data acquisition techniques sub section should have a letter ‘a’ instead of ‘1’

Author response: The subsections in section 4 is now defined as 4a and 4b.

Line 267 - might be worth a sentence on what melanin index and CIELab are (or referencing the relevant figure or section below)

Author response: We have cross referenced section 4b in this sentence.

Line 278 - might be worth defining higher up what you refer to as melanometers vs colorimeters etc

Author response: We have revised Table 1 to include device type – narrowband reflectance, tristimulus colorimeter and reflectance spectroscopy – to define the different types of melanometers.

Line 360 - typo Figure 111

Line 373 - typo figure 122

Author response: The error in Figure numbers has been corrected.

Line 446 - typo “Methos”

Author response: The typo has been corrected.

Line 458 - ?"Error Reference source not found?"

Author response: Table 2 has been correctly referenced.

Line 463 - might qualify that orthostatics dont change hemoglobin concentration of the blood, but blood volume of the tissue which effects hemoglobin content of the tissue volume

Author response: We have reworded the statement as follows (lines 431-432):

"Changes in tissue blood volume during orthostasis or application of a pressure cuff have also been shown to affect objective melanin metrics."

Line 468 - would 'color' be a better word that 'pigmentation' here

Author response: The suggested changes has been made to the statement.

Reviewer #4 (Remarks to the Author):

Vasudevan et al. present a review of melanometry for the assessment of skin pigmentation as a means to improve pulse oximetry. The review is comprehensive and makes extensive use of the literature. This reviewer feels that some major structural changes would make the manuscript stronger, and recommends that the authors strongly consider the proposed edits:

Author response: Thank you. Please find responses below.

1) Greatly shorten sections 5-8. In sections 5-6, the authors presented an analysis of correlations observed in different manuscripts with Fitzpatrick skin phototype and inter-device comparisons, but then the authors ended each section describing why the analysis is flawed overall. This was frustrating to read. The authors should just state the limitations of these types of analyses and move on to the gold standard comparison, which should really be the focus.

Author response: While we agree that this may not be an easy read, this approach was taken in part because flaws have been identified in almost every pigmentation assessment approach, and some of the most flawed, like Fitzpatrick, are still used. Furthermore, knowledge of these flaws should help as the field attempts to identify new approaches (subjective and objective), that hopefully avoid known shortcomings of prior methods. That said, we have attempted to shorten these sections while retaining some of the primary points.

2) It is unclear how the authors interpret the data from papers and the various correlation coefficients. Many different measurements and correlation coefficients are provided, some being quite high, but then no critical analysis is provided on recommendations based on the papers that were reviewed. What are the authors' criteria for an acceptable melanometer? This manuscript would be strengthened considerably by the authors presenting a critique based on the reviewed data, instead of just recommending what needs to be done.

Author response: We have made some critical evaluation based on compiled correlation coefficients and repeatability data. Some examples include:

"Mexameter (MX16, MX18) [ref], Deraspectrometer (DSM) [ref] and Minolta Chromameter (CR200, CR300) [ref] were the most frequently correlated devices with other commercial melanometer outputs.

Even metrics from different measurement techniques (i.e., colorimetry and narrowband reflectance) were well correlated ($|R| = 0.56 - 0.98$) ...”.

“Prior review studies have determined the Minolta Chromameter (CR200/CR300) to be a reliable device for skin color assessment due to its good intra-observer, inter-observer and inter-instrument repeatability compared to other devices [ref]. However, these papers included a much smaller compilation dataset of melanometer reliability studies than what we present here ...”

The above statement indicates that the larger compiled repeatability dataset presented in this paper demonstrates high repeatability in several commercial devices.

“... a high degree of correlation was shown in 85.7% (6 of 7) metric comparisons in commercial melanometers ($|R| > 0.70$). These results indicate that melanometers may often produce higher erythema readings in darkly pigmented skin without any physiological rationale to support this finding.”

“Amongst the common objective melanin metrics (MI, L^* and ITA), 44% of L^* and 77% of ITA results had $|R|$ values above 0.75, with ITA being the most commonly evaluated metric against a gold standard.”

Please note that, considering the limited data available and variability in the compiled results, concluding on one/ a few commercial devices as acceptable melanometers does not seem feasible. Therefore, we signify the importance of standardization and improvement of accuracy in melanometers in our conclusions:

“... standardize melanometry through the establishment of best practices for patient measurements, as well as benchtop and in vivo validation of device performance... advance the accuracy of melanometry through improved instrument design and algorithms for reflectance-based devices, and development of alternate optical sensing technologies.”

3) In general, the Figure captions should contain more text to make them more self contained. There is a lot of information contained in each of the figures, and the captions are too terse to allow a reader to understand what is being shown in the figures.

Author response: We have made reworded the Figure captions to make them more self-contained.

Minor comments

1) p.3, section 2, first paragraph - the authors write "the epidermis, which contains melanin but minimal blood (Figure 1) [53, 70]. The dermis layer is highly vascularized with arterioles and venules but contains minimal melanin".

Comment: The epidermis is avascular, and the (healthy) dermis does not contain any melanin. Please edit the text.

Author response: We have reworded the text as follows (lines 99-101):

“Light delivered to the skin first interacts with the epidermis, an avascular structure containing melanin.”

“The deeper dermis layer is highly vascularized but contains no melanin”

2) p.3, action 2, first paragraph - the authors write "The thickness of epidermis and dermis layers vary strongly from site to site but tend to be in 50 – 150 μm and 1 – 4 mm ranges, respectively."

Comment: Please replace "tend" with "tends" and insert the word "the" between "in" and "50"

Author response: Reworded sentence (lines 101-103) - Epidermis and dermis layer thickness vary strongly with anatomical site but are typically 50 – 150 μm and 1 – 4 mm, respectively

3) p.3, section 2, first paragraph - There is no mention of the stratum corneum in this paragraph. This is an important layer that should at least be mentioned here.

Author response: Considering the length of the manuscript we are not discussing any epidermal and/or dermal layers. Discussion of epidermis and dermis in section 2, paragraph 1 is restricted to the major absorbers and the thickness of these tissue regions.

4) Figure 2 caption - What is YJBM? Is use of this figure in compliance with copyright permissions? In general, please verify that the replicated figures are used in compliance with copyright permissions.

Author response: YJBM - Yale Journal of Biology and Medicine. The requests to reuse figures have been either granted by journal editorial board/authors or have been purchased via CCC RightsLink.

5) p.4 - Please indicate what is meant by a "supranuclear cap"

Author response: Considering the length of the manuscript and lack of relevance of this section to the overall paper, this content has been removed from the manuscript.

6) p.4 - the use of the term "decay rate" is not standard when describing a spectral dependence. Please consider use of a different term.

Author response: We have reworded "absorption decay rate" to "absorption spectra".

7) p.5 - the authors write "the reduced scattering coefficient of melanin pigments also decrease"

Comment: Please replace "decrease" with "decreases"

Author response: The correction has been made.

8) p.5 - the authors mention the "spectral range of pulse oximeters"

Comment: Please provide a wavelength range in the text

Author response: The pulse oximeter wavelengths have been specified in the text.

9) Figure 4 - the acronym SO₂ is not described in the text

Author response: SpO₂ has been defined as oxygen saturation and SaO₂ has been defined as arterial oxygen saturation.

10) Figure 5 - Reflectance values are inconsistent between the two graphs. This reviewer suggests that you change one of the axes to be more similar to the other. Also, are these spectra representative ones or averages? This should be indicated in the figure caption. Finally, the acronym "FST" is used in the figure captions, but "FSP" is used in the text

Author response: The reflectance spectra were collected by broadband reflectance spectroscopy devices (Xrite MA68II and Fieldspec3) on subjects of different FSP types. The figure axes have been modified and the figure legend/captions reflects FSP. Please find modified figure below:

Figure 4 Relationship between tissue optical properties and detected reflectance signals, as illustrated by (a) absorption coefficient spectra for low ($M_r=0.02$, $C_{H_2O}=0.2$) and high ($M_r=0.43$, $C_{H_2O}=0.2$) pigmented epidermis and oxygenated and deoxygenated blood at 150 g/L hemoglobin concentration [ref] and (b) reflectance spectra acquired in different studies with subjects having FSPs I–VI [ref] and FSPs I/II, III/IV, and V/VI [ref]

11) p.6 - replace "FST" with "FSP"

Author response: The correction has been made.

12) Figure 6 - The rightmost figure should be labeled as (b), not (a). This caption would benefit from expanded text that walks the reader through the various graphics.

Author response: The correction to the figure labels have been made. The figures have been described in the text where the figure is cross referenced.

13) p.9 - "HbO₂" and "Hb" have not yet been defined in the text

Author response: HbO₂ and HHb has been defined as oxyhemoglobin and deoxyhemoglobin.

14) p.10 - Please describe what is meant by a "standard observer" and "color matching functions"

Author response: Text has been added as follows (lines 255-257):

"The final type of melanometer discussed is the colorimeter, which quantifies skin color as perceived by the human vision system using the CIE standard observer model, which represents an average human's chromatic response within a 2° field of view."

15) p.10 - Figure 10 is referenced before Figure 9.

Author response: We have combined figures 8, 9 and 10 in the revised version.

16) p.22 - last paragraph of Section 10 - the authors describe "high resolution optical imaging approaches".

Comment: These technologies can offer visualization of melanin at exquisite resolutions, but the one limitation is that the characterization is performed at such a small site, which might not be representative of the skin region of interest.

Author response: We have added text to point out limited region of interest in some of the high-resolution optical imaging approaches (lines 576-578):

"While **restricted region of interest for characterization of melanin** as well as cost and portability of advanced technologies may be a limiting factor to wide acceptance, progress in these technologies will likely lead to more practical options in the future."

Reviewer #5 (Remarks to the Author):

The authors have performed a comprehensive review on the measurement of melanin in the skin by melanometry devices. The measurement by these devices is currently of high interest, because of e.g. the reported influence of skin type on pulse oximetry measurements.

The review is addressing most important issues with the measurements of melanin in the skin, and therefore has value for the scientific community. However, due to the extensiveness of the review, I have the impression that the main message of the manuscript is obscured.

Author response: Thanks. We have attempted to shorten the manuscript and emphasize the sections that are more significant.

1. One of the important factors in the measurement of the melanin content in the skin is the fact that it is non uniformly distributed. It is located in the thin epidermis which make the assessment of e.g. the concentration cumbersome. With visible wavelengths the influence of the underlying skin will influence the outcome of the measurement. In your reference 213, a layered model approach is used which facilitates that the influence of melanin on underlying skin measurements can be decor related. To my opinion these two or multi layered approaches are needed to improve the measurement of melanin in the skin.

Author response: We have added text to Section 10 to emphasize the significance of multi-layered models to quantify melanin concentrations more accurately and have cited the reference – Saager et. al. JBO 2015 (lines 563-564).

"Multi-layered models can help address some of these shortcomings as they can more accurately isolate layer-specific melanin concentration [ref]."

2. melanin is also non uniformly distributed in the epidermis itself, which also has an influence on the measurement of melanin content in the skin. In reference your 89 this non uniform distribution has been incorporated, resulting in good fits of the measured spectra.

Author response: We have added text to Section 10 to emphasize the need to models incorporating non-uniform distribution of melanin in highly pigmented skin and have cited the reference – Zhang et. al. JBO 2019 (lines 562-566).

"However, more advanced models are required to account for the inhomogeneous distribution of melanin in more heavily pigmented skin [ref]"

3. I miss in the discussion the need for layered and calibrated phantoms. These can be essential to validate device outcomes.

Author response: We had noted the potential utility of phantoms for validating and standardizing melanometers in the two final paragraphs of Section 11.

4. With all the knowledge that you have on this topic, I would also expect a conclusive figure with the optical properties of the epidermis. This information can be very useful for modeling optical monitoring devices by the scientific as well as commercial society.

Author response: While we do believe that this is a worthwhile goal, it would involve adding a new set of literature as well as and analysis/discussion of a range of issues that seem beyond the scope of what is already a very extensive review. We have added absorption spectra (350 – 1000 nm) for low and high pigmented epidermis with melanosome volume fractions 0.02 and 0.43 respectively to Figure 2a and equations used to calculate the spectra. Please find lines 132-141 with details and equations below
 “The epidermal tissue absorption can be calculated as follows [ref]:

$$\mu_{a,epidermis}(\lambda) = \left(M_f \mu_{a,mel}(\lambda) + (1 - M_f) \mu_{a,0}(\lambda) \right) (1 - C_{H_2O}) + C_{H_2O} \mu_{a,H_2O}(\lambda)$$

where, M_f is the mean volume fraction of melanosomes in the epidermis, $\mu_{a,mel}$ is the absorption coefficient of a typical melanosome, $\mu_{a,0}$ is the “baseline” absorption coefficient of epidermal tissue without melanin, C_{H_2O} is concentration of water, μ_{a,H_2O} is the absorption coefficient of water and λ is wavelength in nm. The following equations can be used to determine $\mu_{a,mel}$ [ref] and $\mu_{a,0}$ [ref]:

$$\mu_{a,mel} = (519 \text{ cm}^{-1}) \left(\frac{\lambda}{500 \text{ nm}} \right)^{-3.53}$$

$$\mu_{a,0}(\lambda) = 7.84 \times 10^7 \lambda^{-3.255}$$

Figure 5 Relationship between tissue optical properties and detected reflectance signals, as illustrated by (a) absorption coefficient spectra for low ($M_f=0.02$, $C_{H_2O}=0.2$) and high ($M_f=0.43$, $C_{H_2O}=0.2$) pigmented epidermis and oxygenated and deoxygenated blood at 150 g/L hemoglobin concentration [ref]

5. There are some small typos, like reference not found, and figure numbers being 122 and 188.

Author response: The error in Figure numbers has been corrected.

Reviewers' comments:

Reviewer #1 (Remarks to the Author):

The authors have addressed all of my concerns, I think adding a supplemental table (if allowed for a review) that provides additional details around device parameters settings could be useful since this paper will likely be used a reference by many going forward. I appreciate their point about differences in scattering coefficient being 1-10% based on the paper that I shared - however whether that difference is enough to have a significant impact on pulse oximetry is something that remains to be seen and hopefully with further Monte Carlo modeling work will be better understood. Congratulations on an excellent review, thank you for your efforts on this important task.

Reviewer #2 (Remarks to the Author):

The authors have made consistent modification according to the comments of the reviewers. Nevertheless, some issues remain, errors in figures, references (see comments).

Comments:

Line 113 : "Skin color is affected by size, number, and shape of melanosomes, although for the most part the number of melanocytes in a given anatomical site is more consistent with varying pigmentation". This sentence is not clear : the number of melanocytes is rather constant for a given anatomic site between skin of varying pigmentation.

Line 238 : one full stop before "One of the most..." should be deleted.

Line 245: remove space between "...absorption signatures" and ", such as oxyhemoglobin..."

Line 258; "...chromatic response within a 2° field of view". Indeed the observer parameters were standardized as mathematical functions, called 2° and 10° Standard Observers. The 2° Standard Observer represents the average human eye's spectral sensitivity if viewing colors at an arm length distance and from a small field of view, typically used with colorimeters but nowhere in the manuscript is the 10° Standard Observer mentioned, that represents visual assessment from a larger field of view and provides better correlation to human color vision, typically used with spectrophotometers. In general, the manuscript lacks mentioning variables that could affect the device output, therefore the estimates of melanin content and make comparisons between different devices readings more difficult. These include settings such as illuminant, standard observer, colorimetric system, specular component included or excluded, and measurement geometry. This is of the utmost importance and should be included in the Best practices and standardization section.

Line 269 : One space is missing between "...near-infrared reflectance measurements" and [64].

Line 271 : "Since melanin absorption is the primary dermal absorber": the Authors mean "epidermal" not "dermal" as there is no melanin in healthy non pathological skin.

Line 307 : the first publication with the six ITA° groups is the original article ref [167] not the review ref [84].

Line 312; "...HPLC" [169]" add also the reference "5,6-Dihydroxyindole eumelanin content in human skin with varying degrees of constitutive pigmentation". Del Bino S, Ito S, Sok J, Wakamatsu K. Pigment Cell Melanoma Res. 2022 Nov;35(6):622-626.

Line 344: delete space between point and "results".

Line 427: Error! Reference source not found.

Line 474 : one full stop between "[95, 201]" and "Although FM..." should be deleted.

Line 487: "... shows increasing levels of melanosome clusters": this is not correct, Fontana Masson stain is used to demonstrate argentaffinic substance, melanin, non the melanin-producing organelles that are melanosomes. Furthermore, clustered melanosomes are found predominantly in fair skin while dark skin has primarily non aggregated melanosomes

Figure 9 c : The graph showing the levels of eumelanin (PTCA, PDCA) and pheomelanin(TTCA, 4-AHP) markers in different skin color groups is not correct. The graph should be corrected according to Figure 3 a of reference "5,6-Dihydroxyindole eumelanin content in human skin with varying degrees of constitutive pigmentation". Del Bino S, Ito S, Sok J, Wakamatsu K. Pigment Cell Melanoma Res. 2022

Nov;35(6):622-626. that shows contents of 35% of DHI eumelanin (PDCA), 41 % of DHICA eumelanin (PTCA), 20% of benzothiazine pheomelanin (TTCA), and 4% of benzothiazine pheomelanin (4-AHP) regardless the degree of skin pigmentation. The PTCA bar is too high. The scale of the y axis does not correspond. Reference in legend should be modified accordingly.

Figure 9 f and g : looks like it has been drawn from Figure 6 a and b of reference [165] but with less samples. This should be corrected and reference in legend modified accordingly.

Line 515: remove space between "metric" and point.

Line 517: Error! Reference source not found.

Line 652: Include here a paragraph on settings (illuminant, standatd observer, etc).

References: review the references thoroughly to have a homogenous format. Some journals or authors are in capital letters eg ref 18, 21, 28 etc or 143.

if viewing colors at an arm length distance and from a small field of vie

Reviewer #3 (Remarks to the Author):

The authors have addressed the points raised in my initial review and also addressed most points by other reviewers. This will be an incredibly valuable manuscript.

I have only a few minor comments below:

Might consider avoiding the term ethnoracial and more accurately state ethnic or racial. I acknowledge there is variability in preference and reactions with the usage of these terms.

In section 11, might consider adding reference to an ongoing effort/protocol to refine and standardize skin color quantification approaches for pulse oximetry - Open Oximetry Protocol for "Skin color quantification for pulse oximeter human study protocol", 2023, <https://openoximetry.org/study-protocols/>

Adding the aperture sizes of each device in Table 1 would make this table a valuable asset in the literature.

Reviewer #4 (Remarks to the Author):

The authors performed an extremely thorough revision of the manuscript in response to the reviewer comments. This reviewer has no additional comments and is supportive of the manuscript being accepted for publication.

Reviewer #5 (Remarks to the Author):

The authors have improved the manuscript substantially. No further comments.

Thank you for the opportunity to submit a revised draft of the manuscript “Melanometry for Objective Evaluation of Skin Pigmentation in Pulse Oximetry Studies” for publication in the Communications Medicine. We appreciate the time and effort dedicated to providing valuable feedback on our manuscript and thank the reviewers for the insightful comments. We have incorporated most of the suggestions made by the reviewers and any changes have been highlighted within the manuscript. Please find below, a point-by-point response to the reviewer comments and concerns.

Reviewer #1 (Remarks to the Author):

The authors have addressed all of my concerns, I think adding a supplemental table (if allowed for a review) that provides additional details around device parameters settings could be useful since this paper will likely be used a reference by many going forward. I appreciate their point about differences in scattering coefficient being 1-10% based on the paper that I shared - however whether that difference is enough to have a significant impact on pulse oximetry is something that remains to be seen and hopefully with further Monte Carlo modeling work will be better understood. Congratulations on an excellent review, thank you for your efforts on this important task.

Author response: We have included an elaborated version of Table 1 in a supplementary document (Table S1 – Expanded summary of melanometer characteristics).

This table includes columns for device parameter settings: standard illuminants, standard observer, aperture sizes, and inclusion of specular reflectance (with illumination and viewing angles, where available).

We look forward to reading more about this reviewer’s computational findings regarding the effect of epidermal scattering coefficient on pulse oximeter performance.

Reviewer #2 (Remarks to the Author):

The authors have made consistent modification according to the comments of the reviewers. Nevertheless, some issues remain, errors in figures, references (see comments).

Comments:

Line 113 : “Skin color is affected by size, number, and shape of melanosomes, although for the most part the number of melanocytes in a given anatomical site is more consistent with varying pigmentation”.

This sentence is not clear : the number of melanocytes is rather constant for a given anatomic site between skin of varying pigmentation.

Author response: We have clarified the sentence in Lines 113-116.

“Skin color is affected by the type of melanin produced in melanosomes, total melanin content as well as the size, number, shape, and packaging of melanosomes, although the number of melanocytes is rather constant for a given anatomic site regardless of skin pigmentation”

Line 238 : one full stop before “One of the most...” should be deleted.

Author response: This has been rectified.

Line 245: remove space between “...absorption signatures” and “, such as oxyhemoglobin...”

Author response: This has been rectified.

Line 258; “...chromatic response within a 2° field of view”. Indeed the observer parameters were

standardized as mathematical functions, called 2° and 10° Standard Observers. The 2° Standard Observer represents the average human eye’s spectral sensitivity if viewing colors at an arm length distance and from a small field of view, typically used with colorimeters but nowhere in the manuscript is the 10° Standard Observer mentioned, that represents visual assessment from a larger field of view and provides better correlation to human color vision, typically used with spectrophotometers. In general, the manuscript lacks mentioning variables that could affect the device output, therefore the estimates of melanin content and make comparisons between different devices readings more difficult. These include settings such as illuminant, standard observer, colorimetric system, specular component included or excluded, and measurement geometry. This is of the utmost importance and should be included in the Best practices and standardization section.

Author response: We have included text in Lines 260-266 to define 2° and 10° standard observer parameters and specify the devices where they are typically used.

“The distributions that represent spectral sensitivity to light have been standardized as mathematical functions, namely 2° and 10° standard observers (Figure). The 2° standard observer is typically used with colorimeters and the 10° standard observer is typically used with reflectance spectroscopic melanometers [B. C. K. Ly et. al. J Invest Dermatol 2020]. The 2° standard observer represents the average human eye’s spectral sensitivity if viewing colors at an armlength distance from a small field of view whereas 10° standard observer represents visual assessment from a larger field of view and provides better correlation to human color vision [B. C. K. Ly et. al. J Invest Dermatol 2020].”

We have also modified Figure 4d to include both 2° and 10° standard observer functions. Please see updated figure below:

As noted above, an elaborated version of Table 1 has been included in a supplementary document (Table S1). This table includes columns for device parameter settings: standard illuminants, standard observer, aperture sizes, and inclusion of specular reflectance (with illumination and viewing angles, where available).

Text has been included in the Best Practices and Standardization section to specify how device settings can affect the output as well as inter-device performance. Please see detailed response in later comment.

Line 269 : One space is missing between “...near-infrared reflectance measurements” and [64].

Author response: This has been rectified.

Line 271 : “Since melanin absorption is the primary dermal absorber”: the Authors mean “epidermal” not “dermal “as there is no melanin in healthy non pathological skin.

Author response: Statement has been corrected in Line 279.

“Since melanin absorption is the primary **epidermal** absorber in the 600-700 nm region ..”

Line 307 : the first publication with the six ITA° groups is the original article ref [167] not the review ref [84].

Author response: The reference has been corrected and “Del Bino, S., et al., Pigment cell research, 2006.” has been cited in line 315.

Line 312; “...HPLC” [|169]” add also the reference “5,6-Dihydroxyindole eumelanin content in human skin with varying degrees of constitutive pigmentation”. Del Bino S, Ito S, Sok J, Wakamatsu K. Pigment Cell Melanoma Res. 2022 Nov;35(6):622-626.

Author response: “Del Bino, S., et al., Pigment Cell & Melanoma Research, 2022.” has been added as a reference for HPLC and spectrophotometric quantification of melanin in Lines 320 and 321 respectively.

Line 344: delete space between point and “results”.

Author response: This has been rectified.

Line 427: Error! Reference source not found.

Author response: This has been rectified.

Line 474 : one full stop between “[95, 201]” and “Although FM...” should be deleted.

Author response: This has been rectified.

Line 487: “... shows increasing levels of melanosome clusters”: this is not correct, Fontana Masson stain is used to demonstrate argentaffinic substance, melanin, non the melanin-producing organelles that are melanosomes. Furthermore, clustered melanosomes are found predominantly in fair skin while dark skin has primarily non aggregated melanosomes

Author response: The figure caption has been corrected to read “... shows increasing levels of **melanin**”.

Figure 9 c : The graph showing the levels of eumelanin (PTCA, PDCA) and pheomelanin(TTCA, 4-AHP) markers in different skin color groups is not correct. The graph should be corrected according to Figure 3 a of reference “5,6-Dihydroxyindole eumelanin content in human skin with varying degrees of constitutive pigmentation”. Del Bino S, Ito S, Sok J, Wakamatsu K. Pigment Cell Melanoma Res. 2022 Nov;35(6):622-626. that shows contents of 35% of DHI eumelanin (PDCA), 41 % of DHICA eumelanin (PTCA), 20% of benzothiazine pheomelanin (TTCA), and 4% of benzothiazine pheomelanin (4-AHP) regardless the degree of skin pigmentation. The PTCA bar is too high. The scale of the y axis does not correspond. Reference in legend should be modified accordingly.

Author response: Figure 9 c has been updated to Figure 3 a of reference – [Del Bino S, Ito S, Sok J, Wakamatsu K. Pigment Cell Melanoma Res. 2022] and the following text has been added to Lines 506-516.

“Eumelanin consists of 5,6-dihydroxyindole (DHI) and 5,6-dihydroxyindole-2-carboxylic acid (DHICA) moieties, while pheomelanin consists of benzothiazine (BT) and benzothiazole (BZ) moieties. These melanin monomer units can be quantitatively analyzed through specific degradation products by HPLC, including pyrrole-2,3,5-tricarboxylic acid (PTCA) and pyrrole-2,3-dicarboxylic acid (PDCA) for DHICA and DHI moieties of eumelanin, respectively. Pheomelanin moieties BZ and BT can be analyzed as products thiazole-2,4,5-tricarboxylic acid (TTCA) and 4-amino-3-hydroxyphenylalanine (4-AHP), respectively. Figure 9 b illustrates a typical HPLC chromatogram for dark human epidermis. The contribution of these four moieties in 6 skin color groups (very light to dark) based on ITA classification has been demonstrated in Figure 9 c [Del Bino S, Ito S, Sok J, Wakamatsu K. *Pigment Cell Melanoma Res.* 2022]. This paper also showed that proportion of PDCA (35%), PTCA (41%), TTCA (20%), and 4-AHP (4%) is constant regardless of the degree of skin pigmentation [Del Bino S, Ito S, Sok J, Wakamatsu K. *Pigment Cell Melanoma Res.* 2022].”

Please see updated Figure 9 below:

Figure 9 f and g : looks like it has been drawn from Figure 6 a and b of reference [165] but with less samples. This should be corrected and reference in legend modified accordingly.

Author response: Figure 9 e and f (former Figures 9 f and g) are from ref [169] – S. Ito, S. et. al.

"Improved HPLC conditions to determine eumelanin and pheomelanin contents in biological samples using an ion pair reagent," *Int. J. Mol. Sci.* 21, 5134 (2020), and has been accurately stated in the Figure caption. Please note that the references have been included on the top right of Figures 9 d – f (updated Figures included in response to previous comment) to avoid confusion.

Line 515: remove space between "metric" and point.

Author response: This has been rectified.

Line 517: Error! Reference source not found.

Author response: This has been rectified.

Line 652: Include here a paragraph on settings (illuminant, standard observer, etc).

Author response: An expanded version of Table 1 has been included as supplemental information (Table S1) with standard illuminants, standard observers, aperture sizes, and inclusion of specular reflectance (with illumination and viewing angles, where available). The following paragraph has been included in Lines 677-683 to signify the impact of device parameter settings on output and inter-device performance.

“Colorimeters and reflectance spectroscopic melanometers can be operated in different settings. CIE has reported recommendations on the use of standard illuminants, standard colorimetric observers as well as illuminating and viewing conditions for basic colorimetry [CIE, C., Technical report: colorimetry. Commission Internationale de l’Éclairage Central Bureau. Vienna, Austria., 2004]. It should be noted that device settings such as illuminants, standard observer, specular component inclusion/exclusion, and measurement geometry can impact the device output and the estimated melanin content. Therefore, ideally the same device settings should be used to compare values obtained with different colorimetric and spectrophotometric instruments [B. C. K. Ly et. al. J Invest Dermatol 2020].”

References: review the references thoroughly to have a homogenous format. Some journals or authors are in capital letters eg ref 18, 21, 28 etc or 143.

Author response: The references formatting has been corrected to ensure a homogeneous format.

if viewing colors at an arm length distance and from a small field of view

Reviewer #3 (Remarks to the Author):

The authors have addressed the points raised in my initial review and also addressed most points by other reviewers. This will be an incredibly valuable manuscript.

I have only a few minor comments below:

Might consider avoiding the term ethnoracial and more accurately state ethnic or racial. I acknowledge there is variability in preference and reactions with the usage of these terms.

Author response: Line 125-126 has been updated with the term “ethnic or racial.”

“However, ethnic or racial categories provide only a moderate degree of correlation with epidermal melanin content...”

In section 11, might consider adding reference to an ongoing effort/protocol to refine and standardize skin color quantification approaches for pulse oximetry – Open Oximetry Protocol for “Skin color quantification for pulse oximeter human study protocol”, 2023, <https://openoximetry.org/study-protocols/>

Author response: We have included the following text in Section 11 Lines 625-629 and referenced the Open Oximetry Protocol.

“UCSF’s Open Oximetry project is performing a clinical trial of evaluating pulse oximeter performance in a balanced diverse population where skin pigmentation is quantified using multiple color scales as well as colorimeters [Open Oximetry Protocol for “Skin color quantification for pulse oximeter human study protocol”. 2023.]. This project’s “skin color quantification for pulse oximeter human study protocol” represents an ongoing clinical effort to refine and standardize objective skin color quantification approaches for pulse oximetry.”

Adding the aperture sizes of each device in Table 1 would make this table a valuable asset in the literature.

Author response: An elaborated version of Table 1 has been included in a supplementary document (Table S1). This table includes device parameter settings - standard illuminants, standard observers, aperture sizes, and inclusion of specular reflectance (with illumination and viewing angles).

Reviewer #4 (Remarks to the Author):

The authors performed an extremely thorough revision of the manuscript in response to the reviewer comments. This reviewer has no additional comments and is supportive of the manuscript being accepted for publication.

Reviewer #5 (Remarks to the Author):

The authors have improved the manuscript substantially. No further comments.

REVIEWERS' COMMENTS:

Reviewer #2 (Remarks to the Author):

The authors have improved the manuscript significantly addressing all the reviewer concerns, adding a very complete supplemental table S1 and updating figures. This is a great piece of work. It is now suitable for publication, although one issue with ethnicity still remains in one paragraph:

Line 113: "Skin color is affected by the type of melanin produced in melanosomes, total melanin content as well as the size, number, shape, and packaging of melanosomes, although the number of melanocytes is rather constant for a given anatomic site regardless of skin pigmentation [74, 80, 81]". Start the sentence with "Skin color is affected by the melanin content ..." as it is now well known that all skin colors synthesize eumelanin and pheomelanin in the same ratio and the concept that one skin color would be eumelanogenic and another pheomelanogenic is wrong.

Line 116: "A progressive variation in melanosome size with ethnicity has also been revealed, with melanosomes in Black skin being the largest ($1.44 \pm 0.67 \mu\text{m}^2 \times 10^{-2}$) followed by Asian skin ($1.36 \pm 0.15 \mu\text{m}^2 \times 10^{-2}$) and white skin ($0.94 \pm 0.48 \mu\text{m}^2 \times 10^{-2}$) [74, 75, 77, 82]." If possible (double check with the 74, 75, 77, 82 references) replace 'Black' by 'African' or 'African-American' skin and 'White' by 'European'. When possible, geographic origin is a more acceptable descriptor for skin color than ethnic origin. "A progressive variation in melanosome size with ethnicity has also been revealed" replace "ethnicity" by "ethnic or geographic origin".

Line 119, 221 and 123: same remarks as for line 116.

Line 125: [74]. "However, ethnic or racial categories provide only a moderate degree of correlation with epidermal melanin content [63], and extensive variations in melanin content exists within these groups, including by country [85]" "ethnic or geographic" is more suitable than "ethnic or racial".

Thank you for the opportunity to submit a revised draft of the manuscript “Melanometry for Objective Evaluation of Skin Pigmentation in Pulse Oximetry Studies” for publication in the Communications Medicine. We appreciate the time and effort dedicated to providing valuable feedback on our manuscript and thank the reviewers for the insightful comments. We have incorporated most of the suggestions made by the reviewers and any changes have been highlighted within the manuscript. Please find below, a point-by-point response to the reviewer comments and concerns.

Reviewer #2 (Remarks to the Author):

The authors have improved the manuscript significantly addressing all the reviewer concerns, adding a very complete supplemental table S1 and updating figures. This is a great piece of work. It is now suitable for publication, although one issue with ethnicity still remains in one paragraph:
Line 113: “Skin color is affected by the type of melanin produced in melanosomes, total melanin content as well as the size, number, shape, and packaging of melanosomes, although the number of melanocytes is rather constant for a given anatomic site regardless of skin pigmentation [74, 80, 81] “. Start the sentence with “Skin color is affected by the melanin content ...” as it is now well known that all skin colors synthesize eumelanin and pheomelanin in the same ratio and the concept that one skin color would be eumelanogenic and another pheomelanogenic is wrong.

Author response: We have reworded the sentence as recommended in lines 110-112.

“Skin color is affected by the total melanin content as well as the size, number, shape, and packaging of melanosomes, although the number of melanocytes tends to be constant for a given anatomic site regardless of skin pigmentation [ref].”

Line 116: “A progressive variation in melanosome size with ethnicity has also been revealed, with melanosomes in Black skin being the largest ($1.44 \pm 0.67 \mu\text{m}^2 \times 10^{-2}$) followed by Asian skin ($1.36 \pm 0.15 \mu\text{m}^2 \times 10^{-2}$) and white skin ($0.94 \pm 0.48 \mu\text{m}^2 \times 10^{-2}$) [74, 75, 77, 82].” If possible (double check with the 74, 75, 77, 82 references) replace ‘Black’ by ‘African’ or ‘African-American’ skin and ‘White’ by ‘European’. When possible, geographic origin is a more acceptable descriptor for skin color than ethnic origin. “A progressive variation in melanosome size with ethnicity has also been revealed” replace “ethnicity” by “ethnic or geographic origin”.

Author response: Thank you for the feedback. We double-checked the references and have decided to retain the original language used in the references to avoid conflicts with the original paper and any confusion.

We have replaced “ethnicity” with “ethnic or geographic origin” in line 113.

“A progressive variation in melanosome size with ethnic or geographic origin has also been revealed...”

Line 119, 221 and 123: same remarks as for line 116.

Author response: Thank you for the feedback. We double-checked the references and have decided to retain the original language used in the references to avoid conflicts with the original paper and any confusion.

Line 125: [74]. “However, ethnic or racial categories provide only a moderate degree of correlation with epidermal melanin content [63], and extensive variations in melanin content exists within these groups, including by country [85]” “ethnic or geographic” is more suitable than “ethnic or racial”.

Author response: We have replaced “ethnic or racial” with “ethnic or geographic” in line 121. “However, ethnic or geographic categories provide only a moderate degree of correlation with epidermal melanin content...”